# Dynamical mechanisms of growth-feedback effects on adaptive gene circuits

Ling-Wei Kong[1], Wenjia Shi[2], Xiao-Jun Tian[3], Ying-Cheng Lai[1,4]*

[1]School of Electrical, Computer and Energy Engineering, Arizona State University, Tempe, United States; [2]Department of Physics, Xi'an University of Technology, Xi'an, China; [3]School of Biological and Health Systems Engineering, Arizona State University, Tempe, United States; [4]Department of Physics, Arizona State University, Tempe, United States

*For correspondence:
Ying-Cheng.Lai@asu.edu

Competing interest: The authors declare that no competing interests exist.

## eLife Assessment

The paper presents **valuable** computational findings on how growth feedback affects the performance of synthetic gene circuits designed for adaptive responses. By systematically analyzing over four hundred circuit topologies, the authors provide **solid** evidence for their conclusions on failure mechanisms and design features that enhance robustness against growth dynamics. While the study's significance and rigor are somewhat constrained by its reliance on previously published network topologies, these results are highly relevant for advancing the engineering of gene circuits in various applications.

**Abstract** The successful integration of engineered gene circuits into host cells remains a significant challenge in synthetic biology due to circuit–host interactions, such as growth feedback, where the circuit influences cell growth and vice versa. Understanding the dynamics of circuit failures and identifying topologies resilient to growth feedback are crucial for both fundamental and applied research. Utilizing transcriptional regulation circuits with adaptation as a paradigm, we systematically study more than 400 topological structures and uncover various categories of failures. Three dynamical mechanisms of circuit failures are identified: continuous deformation of the response curve, strengthened or induced oscillations, and sudden switching to coexisting attractors. Our extensive computations also uncover a scaling law between a circuit robustness measure and the strength of growth feedback. Despite the negative effects of growth feedback on the majority of circuit topologies, we identify several circuits that maintain optimal performance as designed, a feature important for applications.

## Introduction

In biomedical science and engineering, artificially designed gene circuits are anticipated to play an ever-increasing role in disease diagnosis and therapy (*Riglar and Silver, 2018*; *Sedighi et al., 2019*; *Xia et al., 2019*). Gene circuits also show great potential in various applications such as microbiome modulation (*Foo et al., 2017*; *Lee et al., 2018*) and biological containment (*Gomaa et al., 2014*; *Caliando and Voigt, 2015*). While most gene circuits are designed to function after they are inserted or embedded into host cells, the interactions between the circuit and the host environment are generally extremely complex and can lead to undesired effects that were not present in the original, isolated circuit (*Tan et al., 2009*; *Ceroni et al., 2015*; *Borkowski et al., 2016*; *Ceroni et al., 2018*; *Darlington*

*et al., 2018a*; *Darlington et al., 2018b*; *Kheir Gouda et al., 2019*; *Zhang et al., 2021*; *Zhang et al., 2020*; *Melendez-Alvarez et al., 2021*). Understanding the interactions and identifying the circuit topological structures that can withstand the interactions and thrive in the host are thus of fundamental importance, requiring interdisciplinary efforts among systems and synthetic biology, metabolic engineering, nonlinear dynamics, and complex systems.

Typical circuit–host interactions include metabolic burden, cell growth, and resource relocation or competition, among which growth feedback is the most common type of circuit–host interaction between the circuit gene expressions and cell growth. More specifically, a synthetic gene circuit embedded in a host cell possesses an intrinsic coupling mechanism: the circuit affects cell growth and the growth in turn modifies the gene expressions in the circuit (*Klumpp et al., 2009*; *Klumpp and Hwa, 2014*; *Boo et al., 2019*; *Scott et al., 2010*; *Ray et al., 2016*) – the so-called growth feedback. Studies have shown that the growth-mediated feedback can endow a synthetic gene circuit with various emergent properties, such as innate growth bistability (*Deris et al., 2013*). For example, a non-cooperative positive autoregulation system, when coupled with growth feedback, gains increased effective cooperativity, thereby resulting in bistability (*Tan et al., 2009*; *Nevozhay et al., 2012*). In another example, toxin cooperativity can be induced in multiple toxin–antitoxin systems by growth-mediated feedback (*Feng et al., 2014*). The number of steady states in one gene circuit also depends on growth feedback and resource availability (*McBride and Del Vecchio, 2020*; *Melendez-Alvarez and Tian, 2022*). In general, growth feedback acts to hamper the forward engineering of the circuit functions by introducing modes of nonmodularity and reducing the predictability of the circuit components in an in vivo context. While various phenomena caused by growth feedback were studied with desirable or undesirable effects on the functions of the gene circuits, a systemic picture is lacking on what effects growth feedback can have on the gene circuits, including failures.

A recent study has revealed that growth feedback can have drastically different effects on congruent circuits with distinct topologies (*Zhang et al., 2020*). In particular, the dynamical behaviors of two bistable synthetic memory circuits were studied: a self-activation switch incorporating positive autoregulation and a toggle switch incorporating double-negative regulatory motifs. It was found that growth-feedback impacts both circuits but with quite different manifestations. For the toggle switch, memory can be retained and the circuit tends to be refractory toward growth feedback. However, for the self-activation switch, growth feedback leads to memory loss. While these results indicate that the circuit topology can play a significant role in the circuit functions when growth feedback is present, they were obtained through two specific circuit topologies. Since a particular function of the gene circuit can often be achieved by a finite set of core topologies, it is of fundamental interest to identify the most robust topologies in response to growth feedback. The so-identified optimal topologies can then be used to construct synthetic gene circuits capable of maintaining the essential functions to meet the design goals under the fluctuating growth conditions of the host cell. A systematic study of the interplay between the gene circuit topology and growth feedback is needed.

Adaptation is an important and widely studied functionality of gene circuits, which is defined as the ability of the system to respond to environmental changes and to return to the basal or near-basal state after some time (*Knox et al., 1986*; *Tyson et al., 2003*; *Friedlander and Brenner, 2009*; *Ferrell, 2016*). Previously, it was found that certain circuits possess biochemical adaptation (*Ma et al., 2009*) if they contain at least one of the two architectural classes: an incoherent feed-forward loop (IFFL) with a proportion node and a negative feedback loop (NFBL) with a buffering node. A number of synthetic gene circuits were proposed or constructed to achieve adaptation (*Kim et al., 2014*; *Briat et al., 2016*; *Aoki et al., 2019*). Quite recently, a design principle for circuits with four genes was uncovered for simultaneously achieving noise attenuation and adaptation: the circuit must have a sequential assembly structure (*Qiao et al., 2019*). However, these existing adaptation studies did not include any growth-feedback mechanism.

In this paper, we conduct a comprehensive computational study to uncover and understand the effects of growth feedback on the gene circuits. Specifically, we focus on a type of transcriptional regulation circuit designed for adaptation. There are 425 possible circuit structures (identified by previous research; *Shi et al., 2017*), and we study all of them to simulate and test their response under different levels of growth feedback. Altogether, $2 \times 10^5$ sets of circuit parameters are randomly sampled for each structure. Our results reveal a vast number of cases where growth feedback has a detrimental effect on circuit function ($1.3 \times 10^5$ cases in total) with varying response curves and

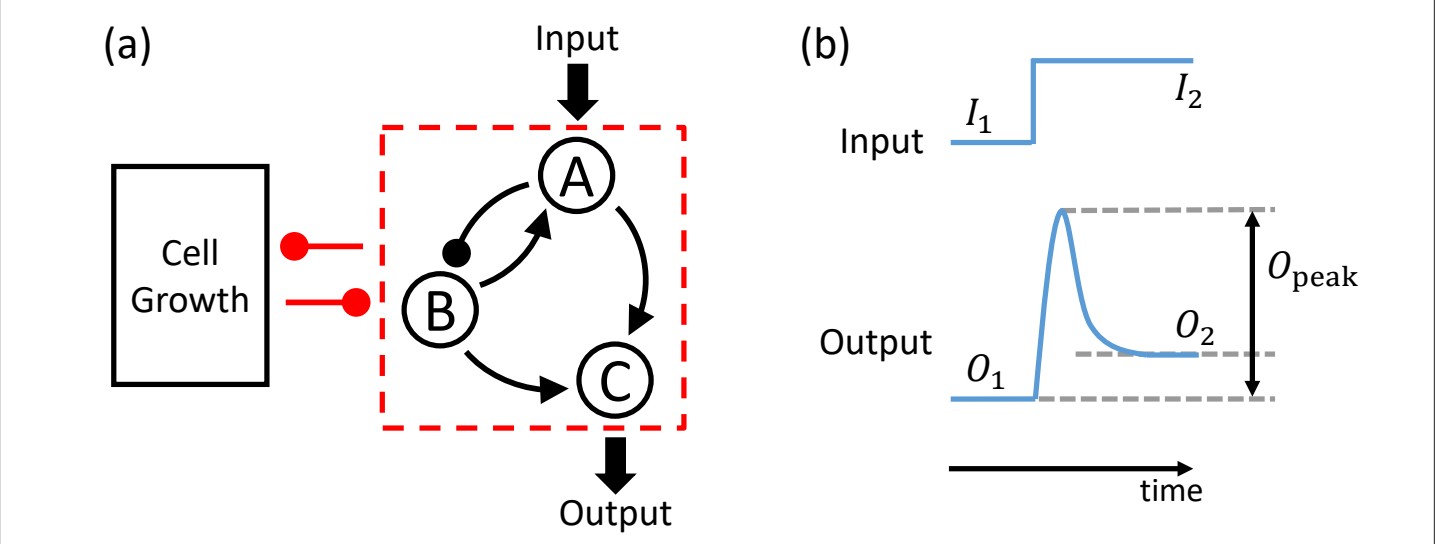

**Figure 1.** Schematic illustration of a synthetic adaptive gene circuit embedded in a host cell. (**a**) A representative three-gene circuit (inside the dashed red box) and its dynamical interplay with host-cell growth. Arrows with triangular ends and round ends denote activating and inhibiting regulations, respectively. Altogether, there are 16,038 possible three-node topologies, with 425 topologies capable of adaptation. (**b**) An example of the circuit input and output signals. The input is an idealized step function of currents $I_1$ and $I_2$ before and after the jump, respectively. The output signal is a response of the circuit to the step function. The features of the output signal, as characterized by three key quantities characterizing the signal: $O_1$, $O_2$, and $O_{peak}$, can be used to determine if the circuit has succeeded or failed in its intended function.

dynamical behaviors. To gain a more intuitive overall picture, we classify these cases into several distinct categories based on the circuits' dynamic behavior. We then systemically summarize the dynamical mechanism behind these growth-induced circuit malfunctions. To quantify circuit adaptation in the presence of growth feedback, we propose a robustness measure that enables us to identify an optimal group of circuits that exhibit a high level of robustness against growth feedback, making them particularly promising for real-world implementation. The motifs associated with this optimal group are found through machine learning. We also obtain a scaling law governing the dependence of this measure on the level of growth feedback and provide a mathematical analysis to gain insights into the underpinnings of the scaling law. The take-home message is that, in spite of the negative effects of growth feedback in the majority of the circuits, there exists a small set of circuits that are still able to deliver optimal performance as designed, which is promising for real-world implementation.

## Results

### A systemic search of functional failures due to growth feedback

Adaptation is referred to as the ability of a gene circuit to respond to changes in input and then to return to the pre-stimulus output level, even when the input change persists (***Ma et al., 2009***). More precisely, with an input signal switched from a lower value $I_1$ to a higher value $I_2$, as demonstrated in ***Figure 1b***, a circuit with functional adaptation should have the following response-curve criteria: (1) precision – the final state $O_2$ should be close to the initial state $O_1$, (2) sensitivity – there should be a relatively high $|O_{peak}|$ in response to the change in the input, and (3) the system should reach equilibrium within a reasonable relaxation time. A three-node gene circuit can achieve adaptation (***Ma et al., 2009***), with one node receiving the input (node A), another node realizing various regulatory roles (node B), and a third node outputting the response (node C). A representative circuit topology is shown inside the red dashed box in ***Figure 1a***. We restrict our study of the class of transcriptional regulatory networks (TRNs) with the AND logic. We fix node A as the input node, and node C as the output node.

Previous research identified 425 different three-node TRN network topologies that can achieve adaptation in the absence of growth feedback (***Shi et al., 2017***), providing the base of our computational study. These topologies can be classified into two families based on the core topology:

networks with an NFBL and networks with an incoherent IFFL (*Shi et al., 2017*). More specifically, there are 206 network topologies in the NFBL family. All of these NFBL topologies have an NFBL for node B. This NFBL can be formed by the loop from node B to A and back to B (such as the circuit shown in *Figure 1a*), by node B to C and back to B, or by a longer route, from node B to A and then to C and back to B. There is always a self-activation link from B to B in all these 206 NFBL networks. There are 219 network topologies in the IFFL family. All of them have two feed-forward pathways from the input node A to the output node C. One pathway goes from node A to C directly, while the other involves node B in the middle. One of the pathways is activating while the other one is inhibitory. We use these 425 network topologies from the study (*Shi et al., 2017*), avoiding redundancy with established results. Due to the unique focus of our research on the effects of growth feedback and the need to evaluate quantitative ratios of robust circuits among all functional ones, we have chosen to use a 20-fold increase in the number of random parameter sets for each network topology compared to the simulations in *Shi et al., 2017*. This approach makes it computationally prohibitive to scan all possible 16,038 three-node circuits. We carefully follow the settings in *Shi et al., 2017*, which also analyzed TRNs with the AND logic as in this paper. Detailed descriptions of our simulation experiments are provided in the Model section. To make our results more convincing, we have adopted a set of adaptation criteria that are stricter than those used in *Shi et al., 2017*. Consequently, the ratio of adaptive circuits is somewhat lower in our study, with 4 out of the 425 network topologies not demonstrating adaptation. The specific structures of these 425 network topologies can be found in our GitHub repository (link provided in Data availability).

In our work, we use a parameter $k_g$ to control the strength of growth feedback, which is a parameter determining the maximal growth rate of the host cells, as mathematically explained in Model description. With all the other parameters fixed, a larger $k_g$ implies a faster cell growth rate and a stronger impact of growth feedback. To investigate the effect of growth feedback on these circuits, we systematically simulate the response of the 425 network topologies under a switch in the input signal. A three-node gene circuit subject to growth feedback has a large number of parameters, which determine the properties of the regulation links within the circuit and the circuit dynamics. For each topology, we randomly sample $2 \times 10^5$ trials of circuit parameters. Altogether, our study involves analyzing approximately $8.5 \times 10^7$ different circuits. We find that among these trials, only about $1.5 \times 10^5$ meet the adaptation criterion in the absence of growth feedback. For these functional trials, we vary the growth-feedback parameter $k_g$ with a series of values, and find that the majority of trials ($1.3 \times 10^5$ trials, about 87%) lose their adaptation in the interval of $k_g \in (0, 1)$, while only 13% of trials remain functional at $k_g = 1.0$.

## A systemic classification of functional failures due to growth feedback

An essential step toward understanding the detrimental or even destructive effects of growth feedback on circuit functioning is to identify the distinct failure scenarios. Our extensive simulations have yielded a comprehensive picture of these scenarios, as shown in *Figure 2*. Overall, we have identified six failure scenarios that encompass more than 99.6% of the $1.3 \times 10^5$ failing cases we collect. The first level of classification distinguishes between failures that occur continuously or abruptly as the growth-feedback strength $k_g$ increases. In a continuous failure, the response curve deforms continuously as $k_g$ increases, as exemplified in *Figure 2a–c*. In an abrupt failure, the response curve exhibits a sudden change as $k_g$ increases through a critical value, as illustrated in *Figure 2d–f*. At the next classification level, we further divide the failures into three types of continuous failures and three types of abrupt failures.

The three types of continuous failures, denoted as types I–III as illustrated in *Figure 2a–c*, are determined according to the specific quantitative criteria in the response curve that the circuits violate. Type-I continuous failures, as shown in *Figure 2a*, are associated with the violation of the precision criterion. A circuit is deemed as precise if a change in the input signal (e.g., from $I_1$ to $I_2$) generates two opposite dynamical effects in the circuit that cancel each other out after a transient and return the final output to the original state, that is $O_2 \approx O_1$. For example, in some networks (e.g., the network in *Figure 1a*), an increase in the input signal $I$ will result in an increase in the concentration of gene $A$ and a reduction in the concentration of gene $B$. As both genes regulate the output gene $C$ with the respective activation links, for proper system parameter values, the two effects will cancel each other out, resulting in $O_2 \approx O_1$. Type-I continuous failures constitute the largest failure category among all

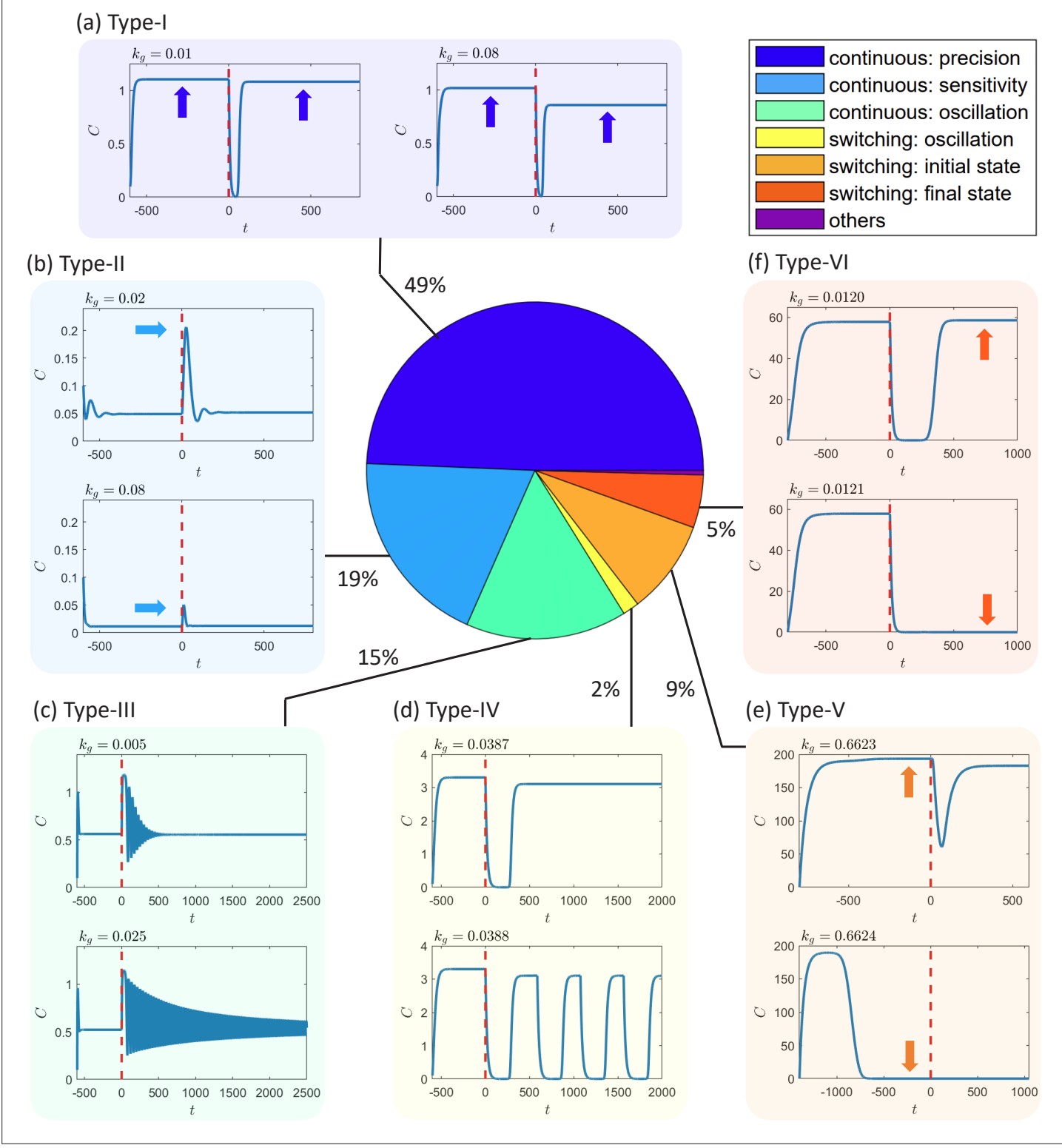

**Figure 2.** Systemic classification of circuit failure scenarios due to growth feedback. This study identifies six computationally detectable categories of failures based on the criterion of functional adaptation that the circuit violates as the effect of growth feedback becomes stronger. (**a**) Type-I and (**b**) type-II failures correspond to the cases where the precision criterion or sensitivity criterion is violated in a continuous fashion as the growth-feedback strength $k_g$ increases, respectively. (**c**) Type-III and (**d**) type-IV failures occur when the circuits lose adaptation due to growth-feedback-induced oscillation, either continuously or abruptly, as $k_g$ increases, respectively. The abrupt changes in type-IV are caused by bifurcations, mostly a saddle-node bifurcation of cycles or an infinite-period bifurcation. For instance, the case shown in (**d**) undergoes an infinite-period bifurcation. (**e**) Type-V and

*Figure 2 continued on next page*

*Figure 2 continued*

(**f**) type-VI failures are when the circuits lose adaptation due to an abrupt change in $O_1$ or $O_2$ as $k_g$ increases, respectively, which are caused by bistability or multistability in the systems. Trials that are not categorized under these six classifications or fall into multiple categories constitute less than 0.4% of all cases (see text for more details and discussions about each failure class). The insets around the pie chart provide exemplary response curves of the circuits in each failure scenario. Each inset shows the concentration of the output node $C$ versus time with two values of the growth-feedback strength $k_g$, one below and another above the failure threshold, for the specific failure scenario. In each case, the input is switched from state $I_1$ to $I_2$ at the time indicated by the red vertical dashed line.

possible circuit topologies, suggesting that the exact cancellation is fragile and the loss of precision is the most common dynamical mechanism behind growth-feedback-induced failures.

Our simulations reveal that an exact cancellation between the two opposite sources at $k_g = 0$ prevents an exact cancellation at any other values of $k_g$. That is, the set of circuit parameter values leading to perfect precision, in general, depends on the value of $k_g$ (see Appendix 5 for more details). The implication is that, for fixed circuit parameter values, achieving high precision under growth feedback ($k_g > 0$) is difficult if the circuit is precise in the absence of growth feedback ($k_g = 0$).

Type-II continuous failures are characterized by a continuous change in the peak of the response curve, denoted as $O_{\text{peak}}$, as $k_g$ increases, eventually falling below a threshold, as shown in *Figure 2b*. This type of failure can make it challenging for downstream circuits to detect the peak signal, hindering information transmission in the larger system. Type-II failures are the second most common type of failure observed in our simulations. The occurrence of a high peak in the response curve requires a significant transient deviation from the final equilibrium point. In the presence of growth feedback, the transient behavior changes, which can further alter the peak height $O_{\text{peak}}$.

Type-III and type-IV failures arise due to growth-feedback-induced oscillations, while type-V and type-VI failures are caused by bistability or multistability. To provide a more detailed understanding of these different failure scenarios, we discuss the two mechanisms, respectively, in the sections of Growth-feedback-induced oscillations and Bistability or multistability.

## Growth-feedback-induced oscillations

As demonstrated by the light green and yellow slices of the pie chart in *Figure 2*, a considerable portion (17%) of the circuit failures are caused by growth-feedback-induced oscillations. Growth-feedback perturbations can easily change the system from the adaptive domain to the oscillation domain in these cases. Our program classifies oscillation-mediated failures into two categories: continuous (type-III) and discontinuous failures (type-IV). Type-III failures are the results of either (1) a gradual increase in the oscillation amplitude or (2) a gradual increase in the transient lifetime of damped oscillations. In the first case, an isolated circuit has already exhibited oscillations with small amplitudes in its gene concentrations with relatively weak growth feedback. As the feedback is strengthened with a larger value of $k_g$, the oscillations are intensified with a larger amplitude, leading to a circuit failure. In the second case, there is damped oscillation for small $k_g$ with a relatively short transient time before approaching an equilibrium. After strengthening the growth feedback, the damping weakens and the oscillation's amplitude cannot be reduced to the threshold within the time limit, as exemplified in *Figure 2c*.

The second category of growth-feedback-induced oscillation is type-IV, where oscillations emerge suddenly as the growth-feedback strength increases through a critical point. The sudden emergence of oscillations can be caused by a bifurcation or a transition into a basin of a limit-cycle attractor. A random sampling of the type-IV failure cases reveals that most of them are caused by either a saddle-node bifurcation of cycles (*Strogatz, 2018*) or an infinite-period bifurcation (*Strogatz, 2018*). In the former case, a pair of stable and unstable limit cycles suddenly emerge together. In the latter case, when observed from the opposite direction (i.e., with a decreasing $k_g$ crossing the threshold), the oscillation in the system spends a longer and longer time around a node on the limit cycle. This node finally becomes a stable fixed point at the bifurcation point, and the oscillation period approaches infinity. One example of type-IV oscillation-mediated failures caused by an infinite-period bifurcation is shown in *Figure 2d*. In our simulations, most of the cases where there are saddle-node bifurcations of cycles are categorized as type-III failures because, prior to the bifurcation point, the system can be oscillating near the ghost cycle (*Strogatz, 2018*) for a long time exceeding the criterion for relaxation

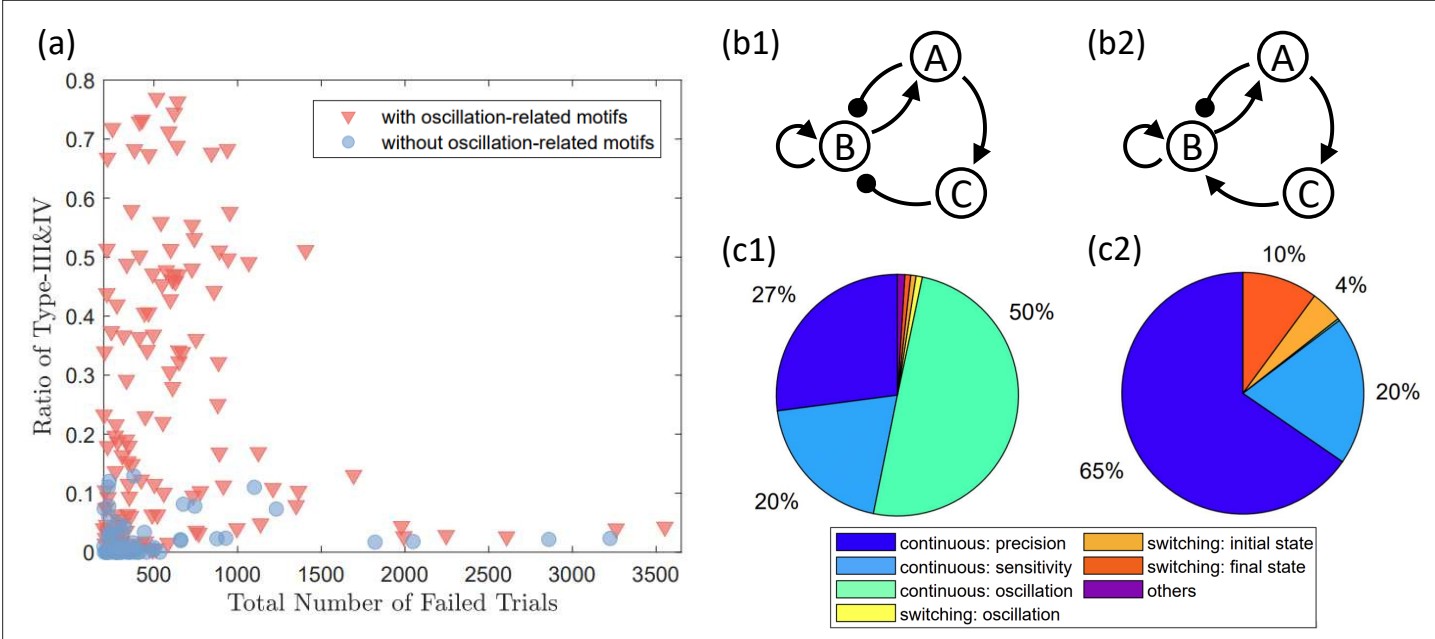

**Figure 3.** Fractions of growth-feedback-induced oscillation failures for different network topologies. (**a**) There are significant variations across network topologies in the fraction of circuit failures attributable to growth-feedback-induced oscillations (types III and IV). Some topologies exhibit virtually no oscillation-related malfunctions, while others experience about 80% of failures caused by growth-induced oscillations. Network topologies containing any oscillation-supporting motifs (discussed in the main text) are represented by red triangles, while the rest are shown as blue circles. The majority of red data points have higher fractions of oscillation-related failures compared to the blue ones, mainly due to the presence of oscillation-supporting motifs. To reduce fluctuations in the results, only circuit topologies with over 200 failed trials are included. (**b1, b2**) A pair of network topologies that differ by only one link (from node C to B). (**c1, c2**) The distinct topologies in (**b1, b2**) leading to different distributions of failure mechanisms. The topology in (**b1**) primarily experiences growth-induced oscillation as the major failure mechanism, while the one in (**b2**) has barely any trials with growth-feedback-induced oscillations.

time, though that ghost cycle is not an attractor but only a transient in the system. More details on these two types of bifurcations in our simulated circuits can be found in Appendix 7.

Our results indicate that for various circuit topologies, the dynamic mechanisms leading to failures can differ, resulting in significantly different fractions of failure types among different networks. For instance, the fractions of failures caused by growth-induced oscillations can vary dramatically among all the topologies, as demonstrated in *Figure 3a*, where each data point represents a specific network topology. The fraction of failures caused by growth-induced oscillations can range from approximately zero to about 80%. A particular example of two different networks is presented in *Figure 3b1, b2*, both of which share the same minimal topology required for adaptation (*Shi et al., 2017*) – the circuit's core function. Despite differing by only one link, the proportions of failures with unique mechanisms are quite distinct, as illustrated in *Figure 3c1, c2*. Notably, for the network in *Figure 3b1*, almost half of the failures result from oscillations, while hardly any oscillation-mediated failures occur for the network in *Figure 3b2*. The explanation is that, although the difference lies in only a single link, this link determines whether an oscillation-correlated motif exists within the network. Previously, three classes of motifs capable of supporting persistent oscillations were discussed (*Novák and Tyson, 2008*), such as the 'delayed negative-feedback loop' in *Figure 3b1*.

Generally, the circuit dynamics depend sensitively on the structure, but oscillations specifically require an NFBL with time delay (*Novák and Tyson, 2008*). Since there are no explicit time-delayed terms in the dynamical equations in our model, one of the two types of motifs – an intermediate node in the path of the NFBL or an additional positive feedback loop – is necessary to induce time delay (*Novák and Tyson, 2008*). For network topologies with a high ratio of functional failures caused by oscillations, both motifs are observed, especially the former type. For the network in *Figure 3b1*, the three links: A → C, C ⊣ B, and B → A, together constitute an NFBL, making the circuit more susceptible to oscillatory behaviors. For the circuit in *Figure 3b2*, no such NFBL exists. *Figure 3a* summarizes the total number of failed trials and the ratio of oscillation-induced failures for each network topology. The

network topologies that contain one of the motifs for oscillation as discussed in *Novák and Tyson, 2008* are marked in red, while the networks that do not consist of any of them are marked in blue. Note that all the networks with relatively high ratios of oscillation-induced failures (e.g., ratio >0.2) include an oscillation-correlated motif. Details about these oscillation-correlated motifs are discussed in Appendix 6.

We conclude that, for a network with an oscillation-correlated motif, even if it is functional at some parameter values, the potential of oscillatory behaviors can be triggered by growth feedback. As a result, networks without these motifs can be safer choices to avoid too many failures cases due to oscillations. Note that this relationship is not absolute. As shown in *Figure 3*, even the networks represented by blue dots that have no oscillation-correlated motifs can still have oscillation-induced failures (with small ratios). The complexity of the scenario makes it challenging to find general and relatively simple rules that connect circuit topology to the circuit's robustness.

## Bistability or multistability

In this section, we describe the dynamical mechanisms behind type-V and type-VI failures, which in total take up about 14% of all the circuit failures. These failures are abrupt, meaning that the response curve undergoes an abrupt change at a certain critical value of $k_g$ from a desirable curve of adaptation. Type-V and type-VI failures correspond to an abrupt change in $O_1$ and $O_2$, respectively. Both types of failures are closely related to bistability or multistability.

Bistability and multistability are common phenomena in nonlinear systems. Bistability refers to the situations where two stable attractors coexist in the phase space simultaneously. Multistability describes a similar coexisting phenomenon of attractors, but with more than two attractors. With bistability or multistability in the target dynamical system, the system trajectory may end up in any one of these stable attractors, depending on the initial state of the system evolution. The entire phase space can thus be separated into two or more basins of attraction. Each basin of attraction corresponds to an attractor and consists of all the initial states that eventually lead the system to the attractor. The boundary boundaries separate two different basins of attraction. A close pair of initial states but at different sides of a basin boundary lead the system to different attractors.

In our simulations, we observe bistability in most of the circuit topologies (377 out of 425 circuit topologies). While multistability has also been observed, it is relatively rare. Considering that both bistability and multistability impact the circuits similarly, our subsequent discussions will primarily focus on bistability, which can be straightforwardly extended to multistability. It is highly unusual for both attracting basins to exhibit the desired functional behavior simultaneously. This is because they are located in different regions of the system phase space, and accommodating both would impose overly stringent constraints on the circuit. After all, having one basin functional is already rare enough with a random sampling of circuit parameters. As a result, functional adaptation is typically found in

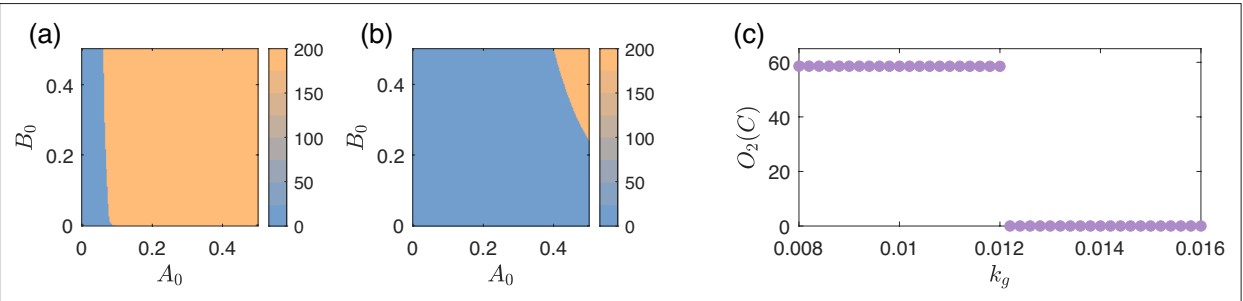

**Figure 4.** Bistability behind both the type-V and type-VI failures. (**a, b**) Basin structure of the circuit shown in *Figure 2e* (with type-V failure) for input $I_1$ with different levels of growth feedback, for growth-feedback strength $k_g = 0.05$ (weak) and $k_g = 0.97$ (relatively strong). The coordinates $A_0$ and $B_0$ are the initial values of nodes A and B, respectively, corresponding to a two-dimensional slice of the entire four-dimensional phase space by fixing $C = 0.1$ and $N = 10^{-3}$. The color bar indicates the equilibrium value of node C before the input switch, which is $O_1(C)$. There is bistability in both cases, as there are two basins of attraction. The yellow region is the functional basin that has adaptation, while the blue region is a non-functional basin without adaptation. The relative size of the blue non-functional region with larger $k_g$ in this case is significantly larger and includes the initial state of the system ($A_0 = B_0 = 0.1$), causing a type-V circuit failure. (**c**) Diagram of $O_2(C)$ from the circuit in *Figure 2f* with a type-VI failure. Prior to a threshold value of $k_g \approx 0.012$, one higher stable value of $O_2(C)$ exists. After this threshold, the state suddenly switches to a lower $O_2(C)$ one. Note that this abrupt change is not caused by a bifurcation. Instead, it is caused by $O_1$ continuously changing with respect to $k_g$ crossing a basin boundary of $O_2$.

only one of the basins, with adaptation being lost in the other, and the circuit is functional only locally in its phase space, rather than on a global scale. A drifting system parameter, such as $k_g$, can alter the dynamics of the gene circuit. In a situation with bistability, such a change in the system dynamics can modify the shape and position of the basin of attraction and the basin boundary. Consider an initial state close to a basin boundary. With the deformation caused by a drifting parameter, the boundary may shift across the initial state, leading to a sudden switching of the system's final attractor. If the basin before the parameter change is functional and the basin after is not, this leads to a growth-feedback-induced failure. The crossing of the basin boundary dictates that the system's final state will abruptly change from one attractor to another. This type of failure can be classified as a switching type of failure.

An example of bistability-related failures is shown in *Figure 2e*, where in the upper panel, the circuit enters into the functional region after an initial transient. In the lower panel, the circuit enters into another region that does not have adaptability, and the circuit does not respond to the switching of the input signal. *Figure 4a, b* illustrates how the basin structure of the circuit changes significantly with different values of $k_g$. The functional basin is in yellow and it shrinks greatly with an increasing $k_g$. Note that the phase space is four-dimensional, so only a two-dimensional slice is shown. For a bistability/multistability-induced type-V failure where $O_1$ is switched, the boundary of the functional basin crosses the initial state. For a type-VI failure, the simultaneous movement of both $O_1$ and the basin boundary under input $I_2$ results in $O_1$ crossing the basin boundary of $O_2$ states.

One might expect bifurcations to play an important role in many type-V and type-VI failures. However, in our simulations, failures precisely at the bifurcation point are not observed. This is because the bifurcation points under consideration, such as fold bifurcations, are where one of the attraction basins diminishes to zero. For a failure to occur exactly at the bifurcation point, the initial condition would need to coincide precisely with the infinitesimally small basin just before it vanishes. More realistically, failures almost always largely precede the exact bifurcation point. They happen while the basin is still contracting and the basin boundary crosses the initial condition or $O_1$. An example is shown in *Figure 4b*, where bistability persists, yet the lighter orange basin with a larger $O_1(C)$ cannot be reached as the boundary shifts away from the initial condition $A_0$ and $B_0$. As another example, in *Figure 4c* from a different circuit, the higher $O_2(C)$ state disappears at $k_g \approx 0.012$ and switches to a lower $O_2(C)$, but this point is not a bifurcation. It is the point where the stable $O_1$ continuously crosses the basin boundary of $O_2$.

## Circuit robustness and optimal topology

To quantify a circuit's robustness against growth feedback, we introduce two metrics: $Q$-value and $R$-value. We track the number of remaining functional trials for each network for various $k_g$ values (starting from $k_g = 0$), denoted as $Q(k_g)$. This measure extends the concept of $Q$-values in *Ma et al., 2009* by accommodating non-zero values of $k_g$. To characterize the circuit robustness, we define

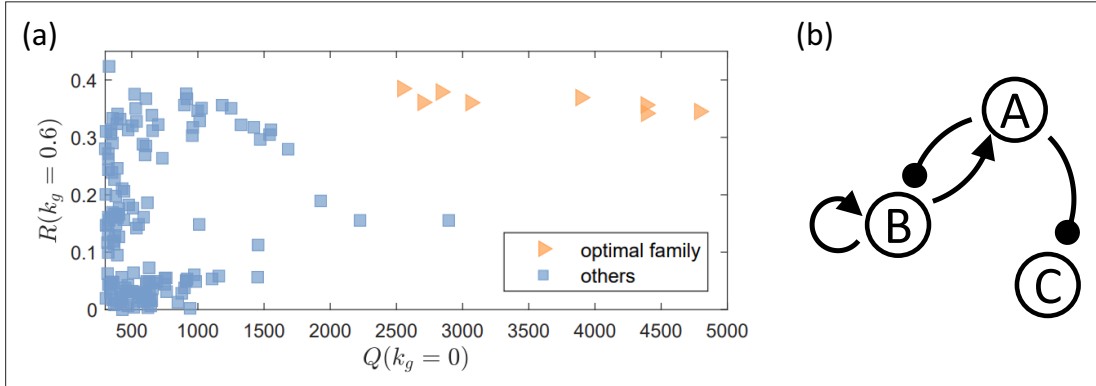

**Figure 5.** A family of circuit topologies with optimal performance. The circuits both have a large volume of the functional region in the parameter space in the absence of growth feedback as characterized by a large value of $Q(k_g = 0)$ and are robust against growth feedback with a high value of $R(k_g)$. (**a**) Values of $Q(k_g = 0)$ and $R(k_g = 0.6)$ from all the 425 network topologies, where each data point corresponds to a topology. The family of optimal topologies is represented by the orange data points, including eight network topologies. (**b**) The set of links (motif) shared by this family of circuits. The combination of these links is also one of the minimal topologies with perfect adaptation in three regulatory logic (*Shi et al., 2017*).

the survival ratio $R(k_g)$ as $R(k_g = k) = Q(k_g = k)/Q(k_g = 0)$. This ratio represents the fraction of random circuit realizations that maintain functionality under growth feedback with a strength of $k_g$.

Note that each $Q(k_g)$ or $R(k_g)$ is defined for a specific network topology in a suitable parameter space. A high value of $R(k_g)$ indicates that a large fraction of the randomly sampled circuit parameters is functional despite cell growth with any strength no larger than $k_g$, indicating that the topology is more robust against growth feedback. Because of the detrimental effects of growth feedback, $R(k_g)$ decreases monotonically with respect to $k_g$.

To justify the utility of $R(k_g)$, we test the circuit topologies employed in a previous work (**Zhang et al., 2020**), where two relatively simple network topologies were used for a comparison study in terms of their ability to resist growth feedback and remain functional. Our evaluation of $R(k_g)$ for the two topologies has yielded results that are consistent with those in **Zhang et al., 2020**, as discussed in Appendix 2. To illustrate our results in a concrete way, we set $k_g = 0.6$ and calculate the ratio $R(k_g = 0.6)$ for different network topologies.

Our computations have revealed a set of eight circuit topologies with optimal performance as characterized by high values of both $Q(k_g = 0)$ and $R(k_g)$, as indicated by the set of orange points in **Figure 5a**. The optimal circuits form a family as their topologies exhibit a high level of similarity with one other. In particular, all eight circuits in this family share a common set of links (motif), as shown in **Figure 5b**. The combination of these common links is one of the minimal topologies with perfect adaptation in a three regulatory logic (**Shi et al., 2017**) and is critical for the circuit to be functionally adaptable. The only difference among the circuits in this family is the links from node C. While an inhibition link from node C can be important to achieving a value of $R(k_g)$, as discussed below, the eight optimal circuit topologies do not contain any such inhibition link from node C. The role of this particular link will be further studied in our analysis of the results in **Figure 6**. This also explains why the family has eight members, as follows. Each link from C has two options: either the link does not appear, or it appears as an activation link. As there are three possible links from C (C to A, C to B, and C to C), there are altogether eight ($2^3$) topologies within this optimal family, according to the simulation results in **Figure 5**.

How can we quickly determine if a three-gene regulatory network with a given topology can be robust against growth feedback? Is there any structural feature of the circuit that can be used to estimate if a high value of $R(k_g)$ can be achieved? To gain insights, we observe that the histogram in **Figure 6b** has three peaks about low, moderate, and relatively high values of $R$, respectively. Computations reveal certain 'shared topological similarity' (or motif) within each peak. Thus, each peak corresponds to a group of similar network topologies that simultaneously have a similar level of $R$. This observation suggests a correlation between the network topology and robustness against growth feedback. For convenience, we refer to these three groups of networks by the colors presented in **Figure 6**. For instance, the group with the highest $R$ the green triangles in **Figure 6a, d** is called the green group, and the group with the lowest $R$ (the red diamonds in **Figure 6a, d**) is the red group.

To better distinguish the three groups, we introduce two binary variables, $B_1$ and $B_2$. For each network, $B_1 = 1$ if the network contains the motif in **Figure 6c**, and $B_1 = 0$ otherwise. Then, for each network, an additional binary variable is set to be $B_2 = 1$ if there is an inhibition link from the output node C, and $B_2 = 0$ otherwise. We find that a linear combination of the two binary variables, $B_S = B_1 - B_2$, can characterize the circuit topology and robustness against growth feedback. In particular, the three possible cases $B_S = 0, 1,$ or $-1$ correspond to the three peaks in **Figure 6b**. This result suggests that the motif shown in **Figure 6c** is beneficial for robustness, while an inhibition link from the output node C is detrimental. It is the balancing act of these two factors that determines the overall circuit robustness.

The discovery of this three-peak structure and the corresponding topological similarity within each peak is facilitated with the use of machine learning. In particular, we consider a simple type of artificial neural network called multilayer perceptron (MLP), where we train it to predict the $R$ value from the input of the network topology through a small hidden layer with only two nodes, as demonstrated in **Figure 6e**. This bottleneck structure in the hidden layer plus the $l-1$ regularization imposed on the input matrix $W_{in}$ forces the MLP to extract low-dimensional features from the input topology to estimate $R$. In our tests, the MLP designed this way automatically assigns different levels of weights to the input information of different links. Over an ensemble of 50 MLPs trained with different random initial

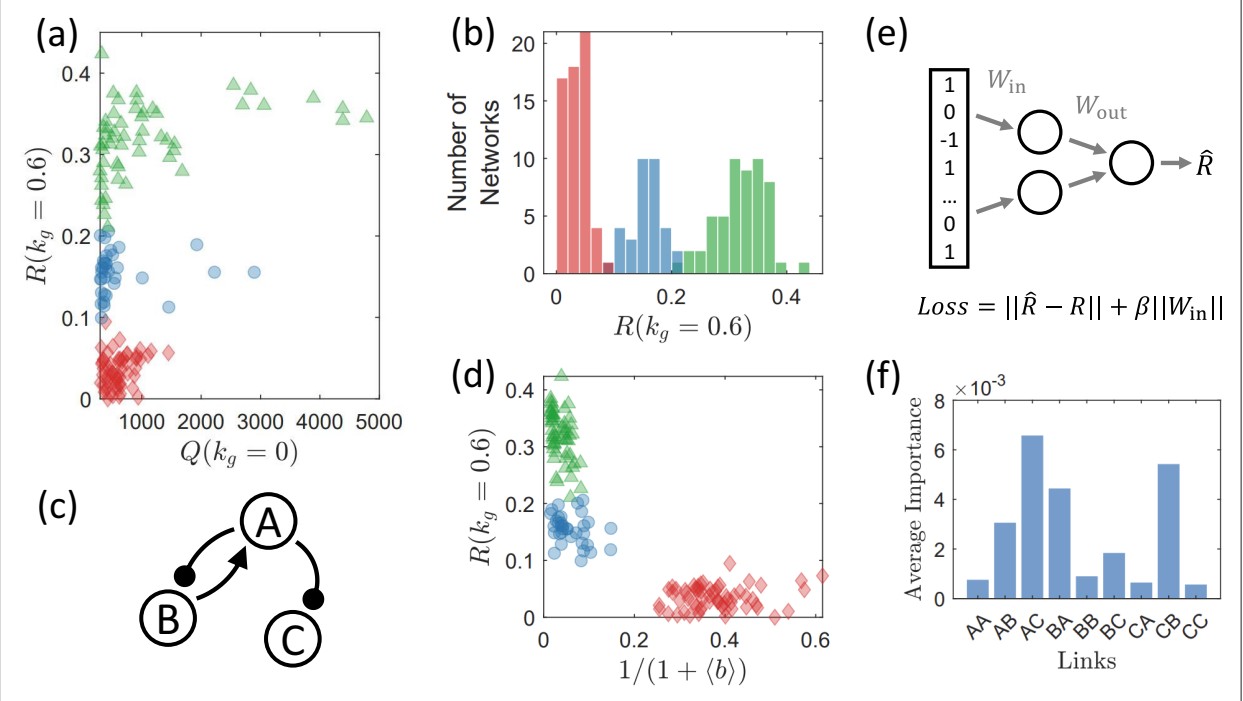

**Figure 6.** Strong correlation between circuit robustness against growth feedback and circuit topology. There are three groups of circuits, each displaying strong topological similarities within, exhibit distinct levels of robustness against growth feedback as measured by the characterizing quantity $R$. (a) Robustness measure $R(k_g = 0.6)$ versus $Q(k_g = 0)$ for all 425 network topologies. Circuits are color/shape-coded into three groups (green triangles, blue circles, and red diamonds) based on the rules defined in the text. The three groups of topologies display distinct levels of $R(k_g = 0.6)$ values, signifying a strong correlation between circuit robustness and topology. Only circuits with $Q(k_g = 0) > 300$ are shown to reduce fluctuations arising from random parameter sampling. What is demonstrated is the case of an intermediate level of growth feedback with $k_g = 0.6$ (a different value of $k_g$ has no significant effect on the results – see **Figure 7**). The topologies associated with the green triangles have a high level of robustness, which can be regarded as an optimal group and is more prevalent than the optimal group identified in **Figure 5**. (b) Histogram of $R(k_g = 0.6)$ the same color legends as in (a). Three distinct peaks emerge, each associated with a group of circuit topologies. (c) The shared network motif among all networks in the green group, which is highly correlated with the optimal minimal network shown in **Figure 5b**, but without the link B → B, which is necessary for the negative feedback loop (NFBL) family of networks to have adaptation (**Shi et al., 2017**). (d) Effects of burden $b$ for the three groups of networks, where the abscissa is the effective term of burden in the formula of growth rate **Equation 12**. The circuits in the red group have larger values of $1/(1 + \langle b \rangle)$, suggesting that a heavier burden yields a stronger effect of the growth feedback for the red group. (e) A multilayer perceptron (MLP) for identifying the crucial connections that determine the robustness of the circuits. The circuit topology serves as the input, where 1, 0, and –1 represent activation, null, and inhibition links, respectively. The output is a predicted robustness measure, denoted as $\hat{R}$. To encourage the neural network to select as few links as possible for predicting $\hat{R}$, a $l$–1 regularization term, $\beta \|W_{in}\|$, is incorporated into the loss function alongside the fidelity error $\|\hat{R} - R\|$. As a result, the feed-forward process eliminates information about the links that have little impact on circuit robustness since the corresponding $W_{in}$ entries automatically optimize to values close to zero. (f) Results from an ensemble of 50 MLPs, each trained with distinct initial values. Shown is the average importance of each of the nine links, which is determined by the weights in $W_{in}$ – see Appendix 8. The top four links with the highest importance correspond to the four links used to classify the three peaks in panel (b).

values, the ranking of average importance is shown in **Figure 6f**. The top four links are the four links used to categorize the three peaks.

The results in **Figure 6** is for $k_g = 0.6$. However, we find that different values of $k_g$ lead to essentially the same ranking of $R(k_g)$ among the circuit topologies, as illustrated in **Figure 7**.

Three remarks on our categorizing rules based on the two extracted featured motifs are in order.

First, the shared motif for the green group is strikingly similar to the optimal minimal network in **Figure 5b** (the orange group). The sole distinction lies in the self-activation link of node B. This specific link plays a crucial role. Every network in the NFBL family depends on this link to achieve adaptation (**Shi et al., 2017**). However, for circuits within the IFFL family, this link is not a necessity for adaptivity. Missing this link makes the motif in **Figure 6c** no longer a minimal network for adaptation, and a circuit containing this motif may either belong to the NFBL or the IFFL family. We have thus identified two optimal groups: the green group with optimal robustness $R$ and the orange group with both the

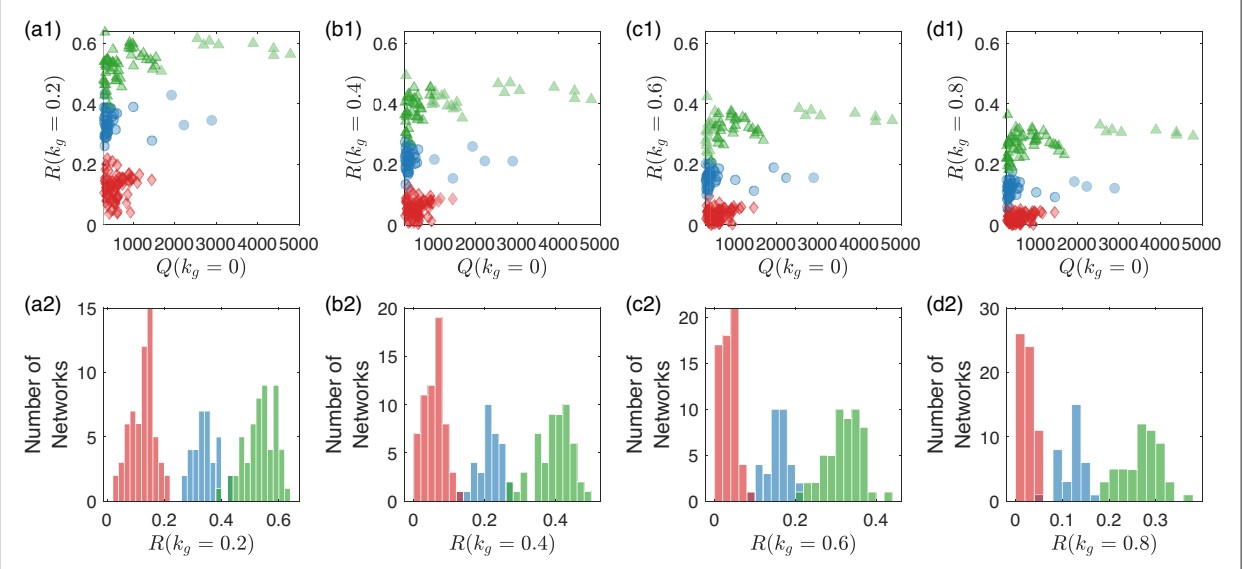

**Figure 7.** Robustness of the circuit division into three groups subject to different levels of growth feedback. From the left to the right, the four columns are for (**a1, a2**) $k_g = 0.2$, (**b1, b2**) $k_g = 0.4$, (**c1, c2**) $k_g = 0.6$, and (**d1, d2**) $k_g = 0.8$, respectively. The legends are the same as in **Figure 6a, b**. For different levels of growth feedback, the distribution of the robustness measure exhibits three distinct peaks that occur at approximately the same locations on the **R** axis. The implication is that the division of the circuit topologies into three groups in terms of the robustness measure can be revealed by examining the circuit functions at a single value of the growth-feedback strength.

optimal robustness $R$ and the largest functional volume $Q(k_g = 0)$ in the absence of growth feedback. The orange group is a subset of the green group, with an additional requirement for $Q(k_g = 0)$.

Second, the shared motif for the red group is also exactly the group of all circuits containing an inhibition link from node C to B, denoted as C ⊣ B. These two different definitions are in fact equivalent: all networks with $Q(k_g = 0) > 300$ that contain C ⊣ A or C ⊣ C also contain the motif in **Figure 6c**, yielding $B_S = 0$, and belong to the blue group.

Third, the three circuit groups in **Figure 6** are not correlated with the categories used in previous research on circuit functionalities without growth feedback (**Ma et al., 2009**; **Shi et al., 2017**). These studies classified adaptive networks into NFBL and IFFL families. Each family contains a few minimal topologies with or without some additional other motifs, and the two families have distinct minimal functional topologies. The minimal topology acts as the backbone for supporting circuit functionality. We find that, when growth feedback is present, the prior classification scheme and the underlying minimal topologies become less relevant. Circuits belonging to the NFBL family are spread across all three levels of $R(k_g)$ in **Figure 6b**, as are the circuits from the IFFL family. A robust circuit can be part of either family, just as a fragile circuit can belong to both. We give that: (1) the topological motifs determining circuit functionality robustness and (2) the motifs deciding whether a circuit belongs to the NFBL or IFFL family are independent. To quantify this irrelevance, we calculate the point biserial correlation between $R(k_g = 0.6)$ and a binary variable determining the family to which the circuit belongs. The resulting correlation is merely 0.1, suggesting hardly any correlations. A further illustration and quantification of this irrelevance can be found in Appendix 4.

What are the reason and mechanism behind the phenomenological set of circuit categories? Especially, it is desired to understand why the shared motif for the green group is beneficial for circuit robustness, and why the shared motif for the red group is harmful for robustness. It is challenging to find straightforward explanations given the complexity of the problem (see Discussion section). Certain insights are as follows. We find that the average node concentrations at the equilibrium for the network topologies in the red group are consistently smaller than those in the blue group. This difference is reflected in the value of burdens $b$. In particular, according to **Equation 12**, the cell growth rate is proportional to the term $1/(1 + b)$ under the same level of growth feedback. **Figure 6d** shows the average burden $\langle b \rangle$ for each network topology, demonstrating that the values of the term $1/(1 + \langle b \rangle)$ for the circuits in the red group are larger than the values in the blue group. As a result,

for the same value of $k_g$, the growth feedback effectively received by the circuits in the red group is stronger than that of the blue group circuits. Further support is provided by the results from the limit $J \to \infty$ (Appendix 3). In this limit, the burden $b$ does not affect the strength of the growth feedback. As a result, the $R$ values of the red group significantly overlap with those of the blue group, suggesting that the distinctively low values of $R$ for the red group be a result of the burden with finite $J$. We also find that the existence of the shared motif for the red group has a stronger correlation to the motif necessary for growth-feedback-induced oscillations. All circuits with oscillation type of failures taking up more than 20% of failures belong to the red group. This correlation can result in further fragility of the red group circuits.

## Scaling law quantifying the effect of growth feedback on gene circuits

A comprehensive way to understand the effects of growth feedback on gene circuits is through scaling laws, an approach commonly employed in statistical and nonlinear physics. Does a scaling law exist that characterizes quantitatively how growth feedback affects the circuit functioning? Through a systematic computational analysis of the circuit robustness, we have uncovered a scaling law that governs how the robustness measure $R(k_g)$ deteriorates as growth feedback is strengthened, as shown in *Figure 8*, where the blue curve is the result averaging over all the 425 network topologies. The three other curves represent circuit topologies that have a relatively high, moderate, and low value $R$ among the 425 topologies tested, to demonstrate that this scaling behavior is generic. These three topologies are the highest $Q(k_g = 0)$ topology in each of the three groups shown in *Figures 6 and 7*. As growth feedback is strengthened, the number of circuit topologies that can maintain functioning decreases (or, equivalently, the number of failed circuits increases). The decreasing behavior of $R(k_g)$ with $k_g$ tends to be slower than exponential [e.g., $\exp(-\beta k_g)$ with $\beta > 0$ being a constant].

A general theoretical argument for the scaling law is unavailable. However, if we simplify the system by setting the parameter $J$ in *Equation 12* to be large so the burden $b(t)$ is much smaller than one, we are able to argue that the scaling law is approximately given by

$$R(k_g) \sim \exp(-\beta k_g^\lambda), \tag{1}$$

where $\beta > 0$ and $0 < \lambda < 1$ are two specific constants that depend on the network topology, and the typical value of $\lambda$ is about 0.6. The exponential scaling is assumed, given its memorylessness. That is, there is no special zero point of $k_g$, for the reason that a certain level of $k_g$ is mathematically equivalent to a larger $d_x$, as discussed below.

The quantity $R(k_g)$ is a simple and straightforward measure characterizing the detrimental effect of growth feedback on gene circuits. We carry out a semi-quantitative analysis of this quantity and the effect of $k_g$ on it. The circuit dynamical equations, *Equations 9–12* can be simplified by substituting *Equation 14* into them to cancel the $dN/dt$ terms, leading to

$$\frac{dA}{dt} = v_A \frac{I^{n_{IA}}}{I^{n_{IA}} + K_{IA}^{n_{IA}}} - (d_A + k_g \frac{1}{1+b})A, \tag{2}$$

$$\frac{dB}{dt} = v_B \frac{A^{n_{AB}}}{A^{n_{AB}} + K_{AB}^{n_{AB}}} - (d_B + k_g \frac{1}{1+b})B, \tag{3}$$

$$\frac{dC}{dt} = v_C \frac{A^{n_{AC}}}{A^{n_{AC}} + K_{AC}^{n_{AC}}} \frac{K_{BC}^{n_{BC}}}{B^{n_{BC}} + K_{BC}^{n_{BC}}}$$
$$- (d_C + k_g \frac{1}{1+b})C. \tag{4}$$

Compared with the equations without growth feedback *Equations 6–8*, we see that introducing growth feedback is equivalent to adding a variable $k_g/(1 + b)$ to the degradation terms for each node. Intuitively, the value $Q$ of a network topology measures the volume of the functional region $\mathcal{M}$ in the parameter space, which is also a function of $k_g$. We thus have that $R(k_g = k)$ is the volume of the intersection between $\mathcal{M}(k_g = k)$ and $\mathcal{M}(k_g = 0)$ divided by the volume of $\mathcal{M}(k_g = 0)$:

$$R(k_g = k)$$
$$= V(\mathcal{M}(k_g = k) \cap \mathcal{M}(k_g = 0))/V(\mathcal{M}(k_g = 0)), \tag{5}$$

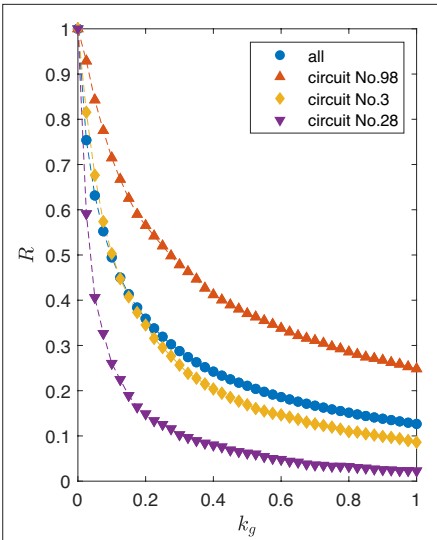

**Figure 8.** Scaling law governing the circuit robustness measure $R(k_g)$. The blue curve is the average result of all the 425 network topologies. The other three curves represent circuits with different robustness levels: high (Circuit No. 98), moderate (Circuit No. 3), and low (Circuit No. 28) values of $R$, to demonstrate that this scaling behavior is generic. Each of these three circuit topologies is selected from one of the three groups illustrated in *Figures 6 and 7*, and they have the highest $Q(k_g = 0)$ value within their respective groups.

where $V(\mathcal{M})$ is the volume of $\mathcal{M}$.

The picture can be further simplified if we assume the burden $b$ is approximately a constant within a range of $k_g$. Since growth feedback contributes to an additional term in degradation $d_x$, strengthening the feedback is equivalent to increasing all three quantities $d_x$ together. Consequently, as $k_g$ increases, the high dimensional region $M$ does not deform, but simply translates in the negative direction in all dimensions of degradation $d_x$ in the parameter space. That is, as growth feedback becomes stronger, it becomes more difficult for the circuit to maintain it functioning.

## Discussion

When a synthetic gene circuit is introduced into a host cell, an inherent coupling arises wherein the gene circuit affects cell growth and cell growth in turn alters the circuit gene expression (*Klumpp et al., 2009*; *Klumpp and Hwa, 2014*). Due to the fundamental nonlinearity in the gene network and in the cell growth dynamics, the interaction is generally quite complicated. To understand this interaction so as to identify the circuit topologies that can withstand the interaction and maintaining the intended circuit functions is one of the most challenging problems in synthetic biology.

Previous studies showed that growth-mediated feedback can endow synthetic gene circuits with various emergent properties. In general, growth feedback tends to negatively impact the intended function the circuit is designed for. There was preliminary evidence that the effects of growth feedback depend strongly on the circuit topology (*Zhang et al., 2020*). For a particular circuit function, while the vast majority of the topologies would fall under growth feedback, a handful still exists that is adaptable to maintain its designed functions.

Identifying the 'optimal' topologies that are most robust against growth feedback is fundamental to constructing synthetic gene circuits that can survive, adapt, and function as designed in the fluctuating growth environment of the host cells.

The main contribution of this paper is a systematic computational study of three-gene circuits with adaptation to uncover and understand the detrimental effects of growth feedback on gene circuits and to identify optimal groups of topologies. Without growth feedback, there are 425 possible topologies with functional adaptation. A vast majority of these circuit topologies fail in their functions under growth feedback, and our computations have revealed, for the first time, six distinct main failure categories covering more than 99% of the cases. From a dynamical point of view, there are three mechanisms by which growth feedback can deprive the circuit of its ability to adapt: (1) continuous deformation of the response curve, (2) strengthened or induced oscillations, and (3) sudden switching to coexisting attractors (also summarized in *Box 1*). By introducing a robustness measure to quantify circuit adaptation in the presence of growth feedback, we uncover a general scaling law characterizing the detrimental effect of growth feedback on the circuit functioning in a quantitative manner. We identify an optimal group of circuits with high robustness and key subsets of links associated with this group that play a critical role in sustaining circuit function in host cells. Taken together, to design a functional gene circuit, growth feedback must be taken into account, as the same circuit designed with perfect functions without the feedback can behave quite differently when the feedback is present. Our study has provided unprecedentedly quantitative insights into the interplay between

## Box 1. Three classes of growth-induced failures.

All the failures we observed can be categorized into the following three general classes, applicable to both the three- and four-gene circuits we tested:

**Continuous deformation of the response curve**

Typically, we require a specific range of response curve shapes for a gene circuit, such as a peak in the output with a minimum height or duration. In a failure caused by continuous deformation, the growth feedback prompts a gradual change that crosses the boundary of these criteria for response curve shapes.

**Growth-induced or growth-strengthened oscillations**

Growth feedback can induce oscillations in a circuit through various types of bifurcations or amplify existing oscillatory behavior with longer relaxation times or larger amplitudes. A circuit experiencing growth-induced or growth-strengthened oscillations cannot reach a relatively steady state (an equilibrium or relatively small oscillations) within a finite time or reasonable relaxation period.

**Growth-induced switching among coexisting attractors**

When coexisting attractors are present in the circuit dynamics, such as bistability or multistability, the circuit typically only functions with one of the attractors. In other words, the circuit is functional locally in its phase space rather than globally. Strengthened growth can push the system across the boundary of different attracting basins in the circuit phase space, causing the circuit to lose its desired functionality by switching from a functional basin to a malfunctioning basin.

gene circuit topology and growth feedback, unlocking the dynamical mechanism of growth-induced failures and providing guidance to better design practically applicable synthetic gene circuits.

A unique finding is that growth feedback can induce or strengthen oscillations in gene circuits designed for adaptation. Such oscillations can often destroy the circuit functionality. In a recent experimental study, a similar phenomenon was observed in gene circuits designed for self-activation (*Melendez-Alvarez et al., 2021*). These results suggest that growth-feedback-induced oscillation may be a general dynamical mechanism that can negatively affect the robustness of gene circuits. In addition, our study has shown that growth feedback has a highly sensitive dependence on the circuit topology: even a small structural differences between two circuits designed for the same function can result in drastically different outcomes under growth feedback. For example, *Figure 3* demonstrates that a link critical for an oscillation-supporting motif can significantly affect the robustness of the circuit against growth feedback. It can thus be quite useful to identify failure-related motifs so that they can be avoided when designing a gene circuit.

From a broad point of view, our study has yielded basic insights into the fundamental topology–function relationships in gene circuits. Examples include how circuit topology affects circuit robustness against growth feedback and whether a circuit topology contains motifs supporting a specific type of growth-induced failure, such as oscillation-related malfunctions. However, searching for and understanding the interplay between circuit topology and dynamical behaviors remain to be a challenge, for the following five reasons.

First, the two relevant questions are whether a circuit topology supports adaptation and whether the circuit is robust against growth feedback or is susceptible to a specific type of growth-induced failures. While our study focused on the latter, the former is important. Addressing both questions to identify and analyze all possible scenarios is infeasible at the present, due to the complex parameter space of the circuits. To make our study feasible, we focused on the cases where the circuit satisfies all the requirements for adaptation in the absence of growth feedback. These cases may occupy a small region in the entire parameter space of the circuit. For each circuit topology, the uncovered function failures due to growth feedback are thus limited to relatively small parameter regions. Second, most network topologies studied have dense connections among the three nodes (only about 20% of the

networks have fewer than six connections). As a result, different motifs can overlap with each other, blocking or enhancing the function of each other. The dense connections thus pose a difficulty in identifying the motifs accurately. Third, for a particular class of failures, competition among different failure types may arise. For instance, a circuit with oscillation-supporting motifs may not have a high fraction of oscillation-induced failures because it also contains the motif for bistability, leading to a large fraction of failures due to the bistability-induced malfunctions. Fourth, due to the necessity to set a threshold in the relaxation time, transient behaviors can arise. In many failure cases caused by oscillations, the oscillatory behavior is not stable and the circuit will eventually approach a fixed point. However, time scales should be taken into account. The transient behaviors can make the network topologies without the necessary motif for sustained oscillations exhibit oscillation-induced failures. Fifth, growth feedback acts as additional feedback loops within the circuit, potentially complicating the circuit dynamics and adding more links to the circuit topology. These extra links in the integrated topology might give rise to an oscillation-related motif. However, our simulations have shown that the impact of this additional oscillation motif, introduced by growth feedback, tends to be weak (Appendix 3).

Although the primary focus of this paper is on how growth feedback can undermine an originally adaptive circuit and how to design circuits that are robust against such feedback, our simulated dataset reveals instances where growth feedback can benefit the circuit within certain ranges. Specifically, we identified 2092 circuits across 306 different topologies where adaption, lost at an intermediate level of growth feedback, is restored at higher levels. This is 1.4% of all circuits tested. We anticipate that additional circuits exhibiting this loss-and-recovery behavior exist, as our sampling of six discrete levels of $k_g$ (0, 0.2, 0.4, 0.6, 0.8, 1.0) might have overlooked numerous cases. This result again suggests the possible advantages of growth feedback in gene circuits (*Tan et al., 2009*; *Nevozhay et al., 2012*; *Deris et al., 2013*; *Feng et al., 2014*; *Melendez-Alvarez and Tian, 2022*). A comprehensive study into how growth feedback can endow or enhance adaption in circuits would require entirely different approaches for sampling circuit parameters and selecting candidate network topologies, demanding significantly high computational costs. Given that this topic extends beyond the scope of the current paper, we leave this matter to future research.

Our study focuses on scenarios where random noises are ignored. Realistically, gene circuits are subjected to diverse types of noise, which can complicate their predictable behavior and design. These noises can originate externally from a noisy input signal $I$, or intrinsically, directly affecting the circuit components. Further, these noises can be classified based on various mechanisms that cause them (*Colin et al., 2017*; *Sartori and Tu, 2011*). And with different mechanisms, each type of noise can be characterized by different attributes such as frequency, amplitude, and noise color. These variances can lead to different impacts on the circuits, potentially necessitating unique mechanisms or designs for the attenuation of each category (*Sartori and Tu, 2011*; *Qiao et al., 2019*). Given the extensive complexity and the need for thorough investigation, these noise-related challenges are beyond the scope of this paper and require a series of future studies.

In our paper, we consider dilution due to cell growth as the dominant factor of growth feedback. Here, we compared the adaptive circuits under no-growth conditions and their ability to maintain their adaptive behaviors after dilution into a fresh medium, which mediated a significant dilution to the circuits. This is based on our previous work *Zhang et al., 2020*. However, growth feedback is inherently complex (*Klumpp et al., 2009*). For instances, an increased growth rate can change protein synthesis rate (*Hintsche and Klumpp, 2013*; *Scott and Hwa, 2023*), and cell growth rates can affect the distribution of protein expression in cell populations (*Kheir Gouda et al., 2019*). In our paper, we concentrate on a simplified model with dilution, which we consider to have captured the dominant factor. The dynamic roles of the dilution and growth-affected production rate should be analogous, given that they both act as inhibitory factors arising from cell growth. Incorporating the impact of growth rate on protein synthesis into our model would offer a more comprehensive analysis, a task beyond the scope of this paper but presenting an intriguing opportunity for future research to address the complexities of growth feedback.

In Appendix 1, we extend our analysis to four-gene circuits with over 2000 functional failure trials. A remarkable finding is that the failure scenarios for these four-gene circuits are the same as the categories for three-gene circuits (summarized in *1*), indicating that the growth-feedback-induced failure mechanisms identified in our work are general. The focus of our study on small gene circuits is

driven by their current relevance in synthetic biology. The primary reason is that even modest sized circuits, when introduced to a host, can provoke unintended and often uncontrollable outcomes due to competition and interactions in the form of growth feedback. Additionally, resource competition within the host cell can arise, where circuit genes compete for limited resources, adversely affecting the dynamics of the circuit. Consequently, larger gene circuits encounter more challenges due to these complexities. Indeed, the state-of-the-art synthetic gene circuits typically involve only three to four genes, a realm where the implications of growth feedback have been insufficiently understood until now. Our work aims to bridge this knowledge gap.

It is possible that, in the future, synthetic biology may use larger and more complex circuits. To uncover and understand the failure mechanisms as well as to identify circuits that are resilient to growth feedback, machine learning can be used. For example, recurrent neural networks have recently been used to identify circuit topologies appropriate for a specified desired function (**Shen et al., 2021**), and reinforcement learning tackle the combinatorial optimization problem (**Bello and Pham, 2016**; **Mazyavkina et al., 2021**) of pinpointing the optimal circuit topologies. Furthermore, automated differentiation (**Hiscock, 2019**; **Kong, 2022**) can be exploited to locate optimal network parameters, which can be efficient for larger circuits with a high-dimensional parameter space. In spite of these works, to study the effects of growth feedback and resource competition among numerous genes in larger circuits remains to be a formidable challenge. Our work providing a comprehensive picture of the failure mechanisms induced by growth feedback represents a step forward in this field.

## Model
### Model description
We restrict our study to the class of TRNs with the AND logic. In order not to overwhelm readers with too many terms and parameters, we first describe a partial model (an isolated circuit without growth feedback) before introducing the complete model that we study in this work. For an isolated circuit without any growth feedback, and with the topology specified inside the red dashed box in **Figure 1a**, the dynamical equations are

$$\frac{dA}{dt} = v_A \frac{I^{n_{IA}}}{I^{n_{IA}} + K_{IA}^{n_{IA}}} \frac{B^{n_{BA}}}{B^{n_{BA}} + K_{BA}^{n_{BA}}} - d_A A, \tag{6}$$

$$\frac{dB}{dt} = v_B \frac{K_{AB}^{n_{AB}}}{A^{n_{AB}} + K_{AB}^{n_{AB}}} - d_B B, \tag{7}$$

$$\frac{dC}{dt} = v_C \frac{A^{n_{AC}}}{A^{n_{AC}} + K_{AC}^{n_{AC}}} \frac{B^{n_{BC}}}{B^{n_{BC}} + K_{BC}^{n_{BC}}} - d_C C, \tag{8}$$

where the dynamical variables $A$, $B$, and $C$ are the concentrations of each protein (node). The notations are as follows. Let $x$ and $y$ be two arbitrary nodes. The quantity $v_x$ is the maximal production rate of gene $x$, $d_x$ is the degradation rate of gene $x$, $dx/dt$ is its time derivative of the concentration, and $n_{xy}$ and $K_{xy}$ are the coefficients in the Hill function for a transcriptional regulation from gene $x$ to $y$.

When growth-mediated feedback is present, the dynamical equations of the three-node circuits are modified to

$$\frac{dA}{dt} = v_A \frac{I^{n_{IA}}}{I^{n_{IA}} + K_{IA}^{n_{IA}}} \frac{B^{n_{BA}}}{B^{n_{BA}} + K_{BA}^{n_{BA}}} - d_A A - \frac{dN}{dt} \frac{1}{N} A, \tag{9}$$

$$\frac{dB}{dt} = v_B \frac{K_{AB}^{n_{AB}}}{A^{n_{AB}} + K_{AB}^{n_{AB}}} - d_B B - \frac{dN}{dt} \frac{1}{N} B, \tag{10}$$

$$\frac{dC}{dt} = v_C \frac{A^{n_{AC}}}{A^{n_{AC}} + K_{AC}^{n_{AC}}} \frac{B^{n_{BC}}}{B^{n_{BC}} + K_{BC}^{n_{BC}}} - d_C C - \frac{dN}{dt} \frac{1}{N} C, \tag{11}$$

$$\frac{dN}{dt} = k_g \frac{1}{1 + b(t)} (1 - \frac{N}{N_0}) N, \tag{12}$$

$$b(t) = \frac{A + B + C}{J}, \tag{13}$$

where the additional dynamical variable $N$ denotes the density of the host cells, $k_g$ is a parameter controlling the maximal growth rate of the host cells, $J$ is a parameter reflecting how this three-node gene circuit contributes to the burden. *Equations 9–13* are the dynamical equations we actually use for simulating the circuit dynamics.

The growth of $N$ is under the regulatory action of two sources: by itself following the logistic equation with the environmental capacity $N_0$ and by the burden $b$ that represents the competence from the metabolism of the gene circuit. To make the computations feasible, we focus our analysis on the exponential growth phase so that $N_0 \gg N$. The equation governing the growth of the cell numbers, *Equation 12*, can then be rewritten as

$$\frac{dN}{dt} = k_g \frac{1}{1 + b(t)} N,\qquad(14)$$

where the dilution rate $dN/dt$ is regulated only by the burden $b(t)$ of the gene circuit. While cell growth is inhibited by the metabolism of the gene circuit, the circuit is also regulated by the growth of $N$ that dilutes the concentration of circuit nodes with increasing cell volume. This dilution is reflected by the additional terms $-(x/N)(dN/dt)$ in *Equations 9–11*.

It is useful to clarify the meaning of the degradation parameter $d_x$ and its relationship to growth feedback. While degradation and growth-feedback terms have the same sign in the regulatory equations, $d_x$ may include a constant dilution. We assume that $d_x$ represents the sum of all the degradation effects in cells that are distinct from growth feedback. For instance, degradation tags, especially in the *ssrA* tagging systems (*Gottesman et al., 1998*), are often used in synthetic gene circuits to increase the degradation rate and thus increase the time scale of the whole system (*Elowitz and Leibler, 2000*; *Fung et al., 2005*; *Stricker et al., 2008*; *O'Brien et al., 2012*).

## Numerical simulations of circuit dynamics

We use the fourth-order Runge–Kutta method to numerically integrate the dynamical equations of the gene circuits, with the integration step as $t_{step} = 0.05$. The dynamical equations we use are similar to *Equations 9–13* but with different topologies. All the initial states of $A$, $B$, and $C$ are taken to be 0.1. The input signal is initially $I_0 = 0.06$ and then switched to $I_1 = 0.6$. The simulation codes can be found in our GitHub repository (link provided in Code availability). The simulation results can be found in our OSF repository (link provided in Data availability).

To achieve the desired adaptation, the circuit's output should reach a steady state before and after the input signal is switched. The values of $O_1$ and $O_2$ can be determined as the output signal associated with the steady states. However, realistically, it is not necessary for the circuit to reach an exact equilibrium. From a computational perspective, the system's state is almost always asymptotically approaching that exact equilibrium but not actually reaching it. As we are simulating the dynamical process of the circuit, setting a condition such as $dA/dt = dB/dt = dC/dt = 0$ to determine $O_1$ or $O_2$ is not meaningful. From a more biological perspective, relatively small drifting or oscillations in the circuit should not harm the circuit functionality and are acceptable. Therefore, we define a 'relatively steady state' where, within a time block of $t_{block} = 200$, the standard deviation of the time series of each node $x(t)$ satisfies: $\text{std}(x) < 1 \times 10^{-4}$ and $\text{std}(x)/\text{mean}(x) < 0.05$. To further guarantee that the circuit is actually in the 'relatively steady state', two successive time blocks satisfying the standard deviation requirements are needed. The quantities $O_1$ or $O_2$ are then defined as the respective mean values of the output signal in that last time block $t_{block}$.

## Numerical criteria for functional adaptation and failure types

We introduce four criteria to determine if a circuit has functional adaptation, and the failure types. The codes implementing these criteria are available in our GitHub repository, with the link provided in Code availability. The failure type results for all circuits tested are available in our OSF repository, with the link provided in Data availability. An additional note is provided in the README file of our GitHub repository for further guidance on generating pie charts similar to *Figure 2* for any network topology or subset of topologies.

## Precision

The basic requirement of adaptation is that the output remains the same when is input is switched from one state to another, that is, $O_2$ should be close to $O_1$ in *Figure 1b*. Specifically, we set the precision criterion to be $|(O_2 - O_1)/O_1| < 0.1$.

## Sensitivity

The circuit is also required to respond to the switch of the input signal with a high peak. This ability of the circuit is named sensitivity. We introduce two types of sensitivity: relative and absolute, with the respective criteria $O_{\text{peak}}/O_1 > 0.5$ and $O_{\text{peak}} > 0.1$. Only the circuits meeting both criteria are regarded as having achieved the required sensitivity.

The need to use the two different criteria simultaneously can be justified, as follows. Given the variety of network topologies and a large number of system parameters, there is a vast diversity in the circuit dynamics and the values of $O_1$. When $O_1$ is small, a peak that satisfies the relative sensitivity criterion alone can still be difficult to observe. If the absolute criterion is used alone for a circuit with a large $O_1$ value, the peak may be negligible in comparison with $O_1$, making its observation practically difficult. It is thus necessary to combine the two criteria so that the cases of small and large values of $O_1$ can be dealt with on the same footing.

## Oscillations and relaxation time

An ideal gene circuit should be able to respond and adapt within a reasonable time scale. We set an upper bound of evolution time $t_{\text{max}} = 4,000$. If the system cannot reach the 'relatively steady state' within this time, it is regarded as non-functional. According to our results, a circuit exceeds our relaxation time upper bound almost always due to oscillations.

## Continuous or abrupt failures

We categorize failures into continuous failures and abrupt switching, as exemplified in *Figure 2*. After determining the circuit's critical $k_g$ value for a circuit, we calculate the changes in $O_1$ and $O_2$ both before and after this critical point to evaluate if the change is continuous or not across the critical value. More specifically, taking $O_1$ as an example, we calculate the $O_1$ values before and after the critical value, $O_{1,\text{before}}$ and $O_{1,\text{after}}$. Then, for each gene A, B, and C, we calculate their changes in $O_{1,\text{before}}$ and $O_{1,\text{after}}$. Taking gene A as an example, if $O_{1,\text{before}}(A) > 0.001$, then a relative difference $|O_{1,\text{after}}(A) - O_{1,\text{before}}(A)|/O_{1,\text{before}}(A) > 0.05$ is considered as abrupt. If $O_{1,\text{before}}(A) < 0.001$, the relative difference is not reliable as when $A$ approaches zero, it can be observed as $A \approx 10^{-10}$ or $A \approx 10^{-12}$, depending on the rather arbitrary time window. So in this case, when $O_{1,\text{before}}(A) < 0.001$, we use a criterion based on the absolute difference $|O_{1,\text{after}}(A) - O_{1,\text{before}}(A)| > 0.01$. If any of the genes have abrupt switching, then the entire failure is considered as an abrupt switching.

## Details of parameter space sampling

A three-node gene circuit subject to growth feedback has a large number of parameters. Let $L$ be the number of links among the three nodes (excluding the input link). The total number of parameters is $2 \cdot 3 + 2(L + 1) = (2L + 8)$. The values of these parameters determine the properties of the regulation links within the circuit and, as a result, the circuit dynamics. The circuit parameters are randomly generated by the Latin hypercube sampling method (*Iman, 1980*) using the function 'lhsdesign' in Matlab. The parameters are sampled uniformly either on a logarithmic or a linear scale. The sampling ranges of the parameters are: $v_x \in [10^{-1}, 10^1]$ (sampled in logarithmic scale), $d_x \in [10^{-2}, 1]$ (sampled in logarithmic scale), $n_{xy} \in [1, 4]$ (sampled in linear scale), and $K_{xy} \in [10^{-3}, 1]$ (sampled in logarithmic scale).

## Code availability

All computer codes can be found at GitHub (copy archived at *Kong, 2025*).

## Acknowledgements

This work was supported by the Air Force Office of Scientific Research under Grant No. FA9550-21-1-0438 (to YCL), by NIH grant R35GM142896 (to XJT), and by the Young Talent Fund of the University Association for Science and Technology in Shaanxi, China, Grant No. 20210506 (to WS).

## Additional information

### Funding

| Funder | Grant reference number | Author |
|---|---|---|
| Air Force Office of Scientific Research | FA9550-21-1-0438 | Ying-Cheng Lai |
| National Institutes of Health | R35GM142896 | Xiao-Jun Tian |
| Young Talent Support Program of Shaanxi Province University | 20210506 | Wenjia Shi |

The funders had no role in study design, data collection, and interpretation, or the decision to submit the work for publication.

### Author contributions

Ling-Wei Kong, Conceptualization, Software, Formal analysis, Validation, Investigation, Visualization, Methodology, Writing – original draft, Writing – review and editing; Wenjia Shi, Formal analysis, Funding acquisition, Validation, Investigation, Methodology; Xiao-Jun Tian, Conceptualization, Formal analysis, Funding acquisition, Validation, Investigation, Methodology, Writing – review and editing; Ying-Cheng Lai, Conceptualization, Formal analysis, Supervision, Funding acquisition, Validation, Investigation, Methodology, Writing – original draft, Writing – review and editing

### Author ORCIDs

Ling-Wei Kong (iD) https://orcid.org/0000-0002-8921-1642
Ying-Cheng Lai (iD) https://orcid.org/0000-0002-0723-733X

Joint Public Review: https://doi.org/10.7554/eLife.89170.3.sa1
Author response https://doi.org/10.7554/eLife.89170.3.sa2

## Additional files

### Supplementary files

MDAR checklist

### Data availability

All relevant data can be found at GitHub (copy archived at *Kong, 2025*) and Open Science Framework.

The following datasets were generated:

| Author(s) | Year | Dataset title | Dataset URL | Database and Identifier |
|---|---|---|---|---|
| Kong LW | 2024 | Effects of growth feedback on adaptive gene circuits: A dynamical understanding | https://doi.org/10.17605/OSF.IO/PZY7R | Open Science Framework, 10.17605/OSF.IO/PZY7R |
| Kong LW | 2025 | Growth_Feedback_Adaptation | https://github.com/lw-kong/Growth_Feedback_Adaptation | GitHub, lw-kong/Growth_Feedback_Adaptation |

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

## Appendix 1

### Four-gene circuits

To demonstrate the general applicability of our nonlinear dynamical analysis of the failure mechanism, we study four-gene circuits. *Appendix 1—figure 1* shows ten representative circuits, where eight are from *Qiao et al., 2019* and two being the four-node modifications of three-gene circuits with oscillation-related motifs. For each circuit, we test $10^5$ random sets of parameters. To generate acceptable statistics, we ease the precision and sensitivity criteria to: (1) $|(O_2 - O_1)/O_1| < 0.4$, (2) $O_{\text{peak}} > 0.1$, and (3) $O_{\text{peak}}/O_1 > 0.5$ or $O_{\text{peak}}/O_1 - |(O_2 - O_1)/O_1| > 0.1$. All other simulation settings are the same as those in the three-gene circuit simulations as detailed in the main text. We collect a total of 3275 trials exhibiting functional adaptation in the absence of growth feedback ($k_g = 0$). As the growth feedback is turned on so that $k_g = 0$ increases $k_g = 0.5$, 2373 trials encountered functional failures across all 10 circuits.

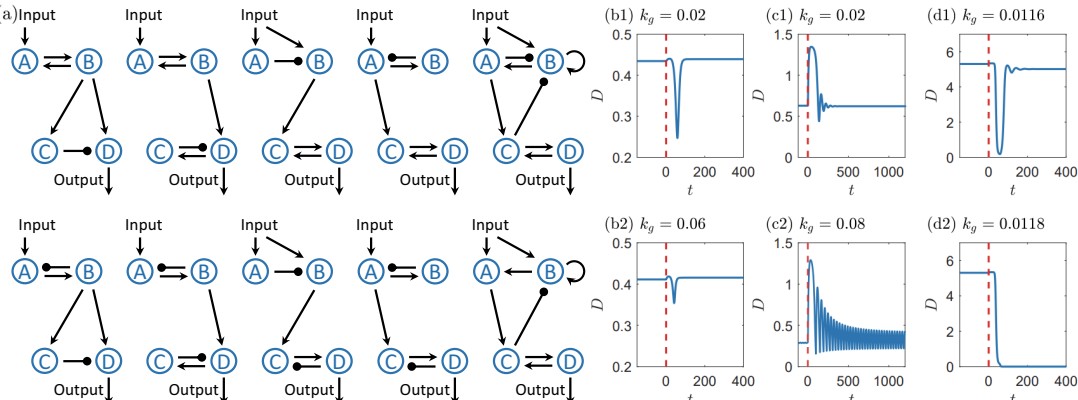

**Appendix 1—figure 1.** Functional failures in four-gene circuits under growth feedback. (**a**) Ten representative four-gene circuits. The eight circuits in the first four columns are from *Qiao et al., 2019*, and the two circuits in the fifth column are selected due to the oscillation-related motifs in their topologies and the relatively high $Q(k_g = 0)$ values when reduced to three-gene circuits. (**b–d**) Examples of the three major categories of growth-feedback-induced functional failures in the four-gene circuits, where the upper panels display the circuit outputs with smaller $k_g$ values for which the circuits remain functional and the lower panels showcase the circuit outputs with larger $k_g$ values for which the circuits lose their functionality. The vertical red dashed line marks the time when the input is switched to another state. The three failure categories are identical to these in the three-gene circuits in the main text: (**b1, b2**) continuous trajectory deformation causing the system to cross thresholds associated with the sensitivity criterion, (**c1, c2**) growth-strengthened oscillations, and (**d1, d2**) growth-induced switching in bistability. The change in $k_g$ between panels (**d1**) and (**d2**) is small so as to show the abrupt change in the response at a critical point.

We then investigate the causes of the functional failures. We find that all 2373 trials fall into the same three categories identified for three-gene circuits: growth-induced oscillations, growth-induced switching in bistability, and continuous deformation of the system trajectory leading the system to cross the criteria thresholds, as shown in *Appendix 1—figure 1b–d*, respectively. For the cases studied, continuous deformation is the dominant failure mechanism, accounting for about 88% of the failures. The fractions of oscillation- and bistability-related failures are approximately 10% and 3%, respectively. These results indicate that four- and three-gene circuits share the common mechanisms of growth-feedback-induced failures, implying the generality of these failure mechanisms.

## Appendix 2

### Self-activation and toggle switch circuits

The key quantitative results about the survival ratio $R(k_g)$ presented in the main text are obtained from various circuit topologies with three genes. To demonstrate the general applicability of $R(k_g)$, we study two simpler gene circuits: a self-activation circuit with a single gene and a toggle switch circuit with two genes. A comparative study of these two classes of circuits has been carried out recently (*Zhang et al., 2020*), whose topological structures are shown in *Appendix 1—figure 1a1, a2*, respectively. In the absence of growth feedback, both networks exhibit bistability and a hysteresis loop. Under dilution, the self-activation circuit quickly loses the memory while the toggle switch circuit can remain functional, as was observed numerically and experimentally (*Zhang et al., 2020*).

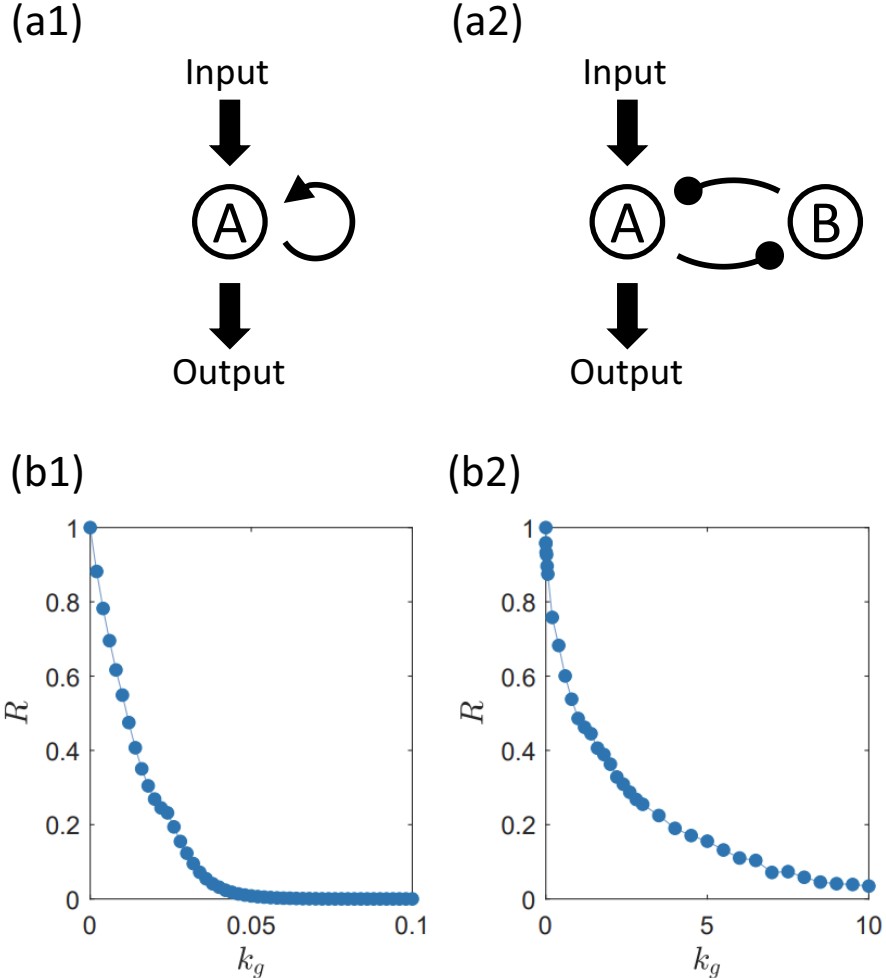

**Appendix 2—figure 1.** Scaling law of robustness measure for the single-gene self-activation circuit and the two-gene toggle switch circuit. (**a1, b1**) The topology of the self-activation circuit and the decay of the robustness measure with the growth-feedback strength. (**a2, b2**) Same legends as (**a1, b1**), respectively, for the toggle switch circuit. Note the drastic difference in the range of $k_g$ values in (**b1**) and (**b2**) where $R$ approaches zero much more quickly in the former than in the latter, indicating the nearly immediate loss of functions of the single-gene circuit even under weak growth feedback.

Our simulation settings are mostly identical to that of three-node circuits in the main text, including the sampling regions of the random circuit parameters, the specifics of the ODE solver, and the criterion for locating equilibrium. We set $J = 1$. Other than the network topology, the only difference is the functionality criteria. Here, the desired function is a hysteresis. We test the response of the circuit output when (1) the input is a switch from an off-state (with input signal $I_{\text{off}} = 10^{-8}$) to an on-state (with input signal $I_{\text{on}} = 2$) and (2) the input is switched from an on-state to an off-state. In the former

trial, the steady-state output is switched from $O_{1,\text{off}}$ to $O_{1,\text{on}}$, while in the latter it is switched from $O_{2,\text{on}}$ to $O_{2,\text{off}}$. The criteria are: (1) the two steady states are distinguishable: $\Delta O = O_{2,\text{on}} - O_{1,\text{off}} > 0.1$; and (2) the system exhibits a hysteresis: $(O_{1,\text{on}} - O_{1,\text{off}})/\Delta O > 0.5 > (O_{2,\text{on}} - O_{2,\text{off}})/\Delta O$.

*Appendix 2—figure 1b1, b2* show the scaling law of $R(k_g)$ with $k_g$ for the self-activation and toggle switch circuits, respectively. It can be seen that, for the self-activation circuit, as the growth-feedback strength increases, $R(k_g)$ approaches zero quickly, indicating that the circuit function cannot sustain even weak feedback with near zero strength. For the toggle switch, $R(k_g)$ approaches zero eventually but at a much slower rate, a result that is consistent with the finding in *Zhang et al., 2020*. Remarkably, the scaling of $R(k_g)$ with $k_g$ exhibits qualitatively similar behavior as the scaling laws reported in the main text for various three-gene circuits, lending further credence for the general applicability of the quantitative measure $R(k_g)$ to characterize the effects of growth feedback on gene networks.

## Appendix 3

### Results from low burden level

For the simulation results reported in the main text, the burden parameter is fixed at $J = 1$. What are the possible behaviors of the gene circuit for different values of $J$? Suppose $J$ is much larger than one. In this case, the burden term $b$ that has $J$ in the denominator is negligible, thereby reducing the complexity of the system and providing a parameter regime in which the contributing factors to the survival ratio $R(k_g)$ other than the burden can be identified.

In the regime of large $J$, the burden in *Equation 8* in the main text is much smaller than one, so *Equation 7* in the main text about growth rate can be simplified as

$$\frac{dN}{dt} = k_g \frac{1}{1+b}N = k_g \frac{1}{1+(A+B+C)/J}N \approx k_g N, \tag{15}$$

indicating that cell growth is determined entirely by the growth-feedback strength $k_g$. It can be seen from *Equations 4–6* in the main text that, in this case, the effect of growth feedback is equivalent to a linear change of the amount $k_g$ in the degradation terms $d_x$. Further, the interaction between cell growth and the gene circuit is no longer of the type of mutual inhibition: the regulation is a one-way interaction from cell growth to the gene circuit. A semi-quantitative analysis of this scenario can be found in Appendix 5.

We carry out the simulations as in the main text in the regime of large $J$ and perform a comparative analysis of the results.

The first issue concerns the relative fractions of different failure scenarios. *Appendix 3—figure 1* compares the distributions of distinct types of circuit failures for $J = 1$ and $J \to \infty$. The possible failure scenarios are identical in both cases, in spite of the quantitative differences in the relative fractions of the failure mechanisms. Some of the differences are sizable, but none is significant in the sense that none is beyond an order of magnitude. For example, for $J = 1$, type-I failures are the most common (49%) where the precision criterion is broken in a continuous fashion. For $J \to \infty$, the fraction is about 31%, but the reduction is still within a factor of two. The plausible reason for the reduction is that the additional regulation of the burden $b$ for $J = 1$ is more difficult to be maintained (Appendix 5).

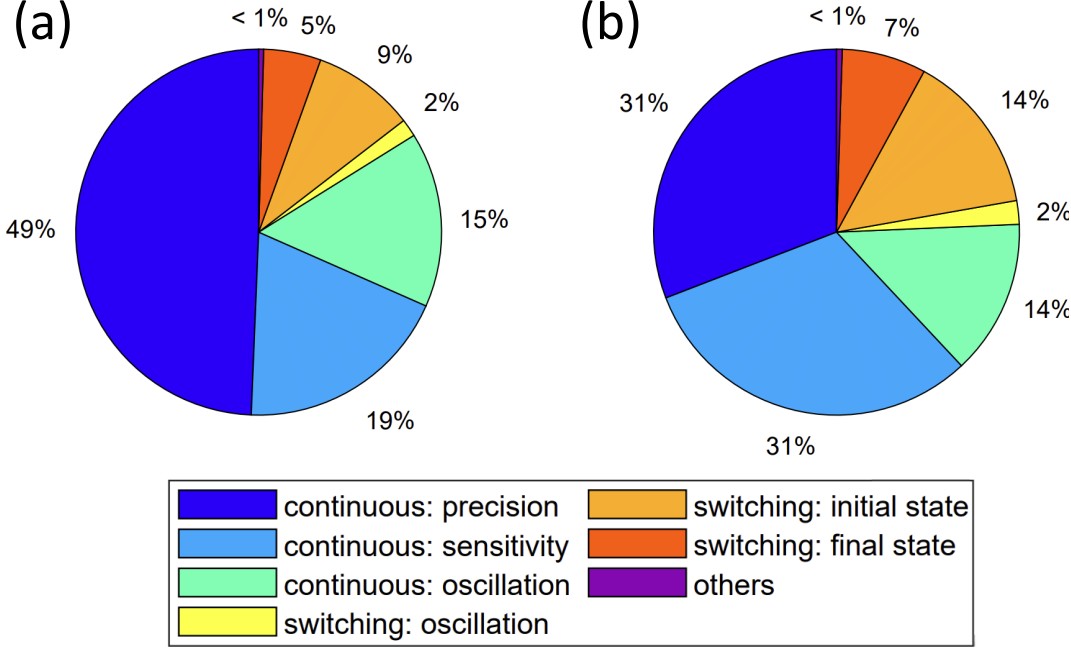

**Appendix 3—figure 1.** Circuit performance for zero burden. Shown is a comparison of the distributions of circuit failure scenarios under growth feedback for (**a**) $J = 1$ as in the main text and (**b**) $J \to \infty$ (zero burden). In both cases, there are six categories in spite of some quantitative differences in their probabilities, implying that, as the burden is reduced to zero from a finite value continuously, the failure scenarios are qualitatively the same. Notable

is the fraction of circuits suffering type-I failures (violation of the precision criterion), which has a relatively large reduction for $J \to \infty$, a result that is consistent with the semi-quantitative analysis in Appendix 5.

The second issue is the scaling law between the survival ratio $R(k_g)$ and the growth-feedback strength $k_g$. *Appendix 3—figure 2* compares the scaling laws of $R(k_g)$ for three circuit topologies for $J = 1$ and $J \to \infty$, where the results in panels (a1) and (a2) are represented on a linear scale, while those in panels (b1) and (b2) are on a double-logarithmic versus logarithmic scale. The approximately linear relation in panel (b2) suggests that, for $J \to \infty$, the scaling laws is given by (1) in the main text.

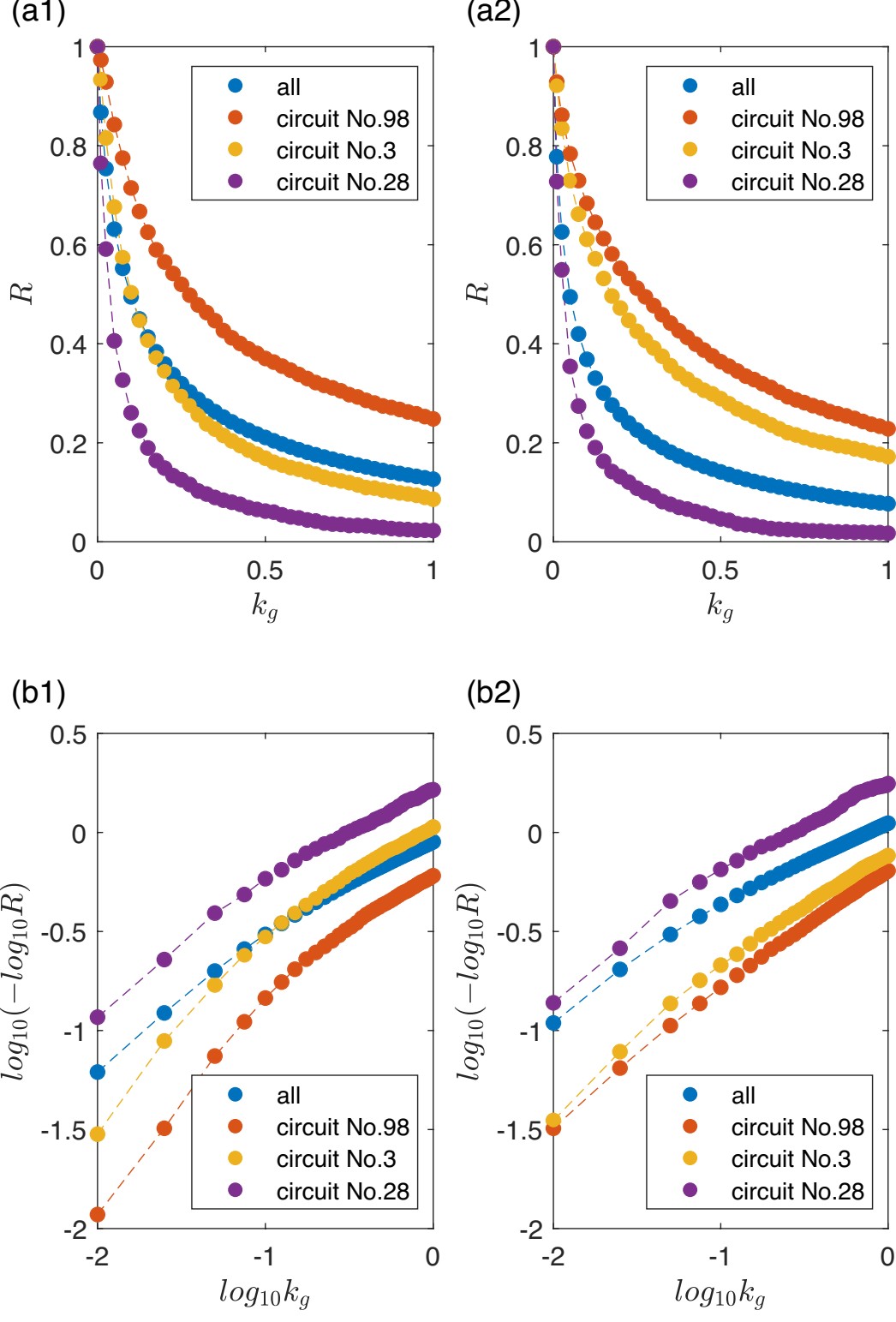

**Appendix 3—figure 2.** Scaling law of circuit robustness measure for zero burdens. (**a1, b1**) Representative scaling relations between $R(k_g)$ and $k_g$ for $J = 1$ as in the main text, plotted on two different scales. (**a2, b2**) Representative scaling relations for $J \to \infty$. The curves in (**b2**) are approximately linear, suggesting the scaling law (1) in the main text. In (**b1**), the curves are less linear where the added burden leads to more reduction in $R(k_g)$ in the regime of weak growth feedback.

For $J = 1$, the scaling law (1) is less accurate, as shown in **Appendix 3—figure 2b1**, which can be heuristically explained, as follows. Suppose we use **Equation 15** and reduce $J$ from a large value to one, which is equivalent to adding back the negative feedback from the burden $b = A + B + C$ to cell growth. Since cell growth effectively inhibits the gene regulation in the circuit, the burden will be larger for smaller values of $k_g$, suppressing the cell growth. Thus, for weak growth feedback (corresponding to small values of $k_g$), for small $J$, $R(k_g)$ decreases more slowly than for larger values of $J$. The difference becomes smaller for larger values of $k_g$, causing the curves on the left side in **Appendix 3—figure 2b1** to be lower than those in **Appendix 3—figure 2b2**, but the curves on the right side are similar in both cases.

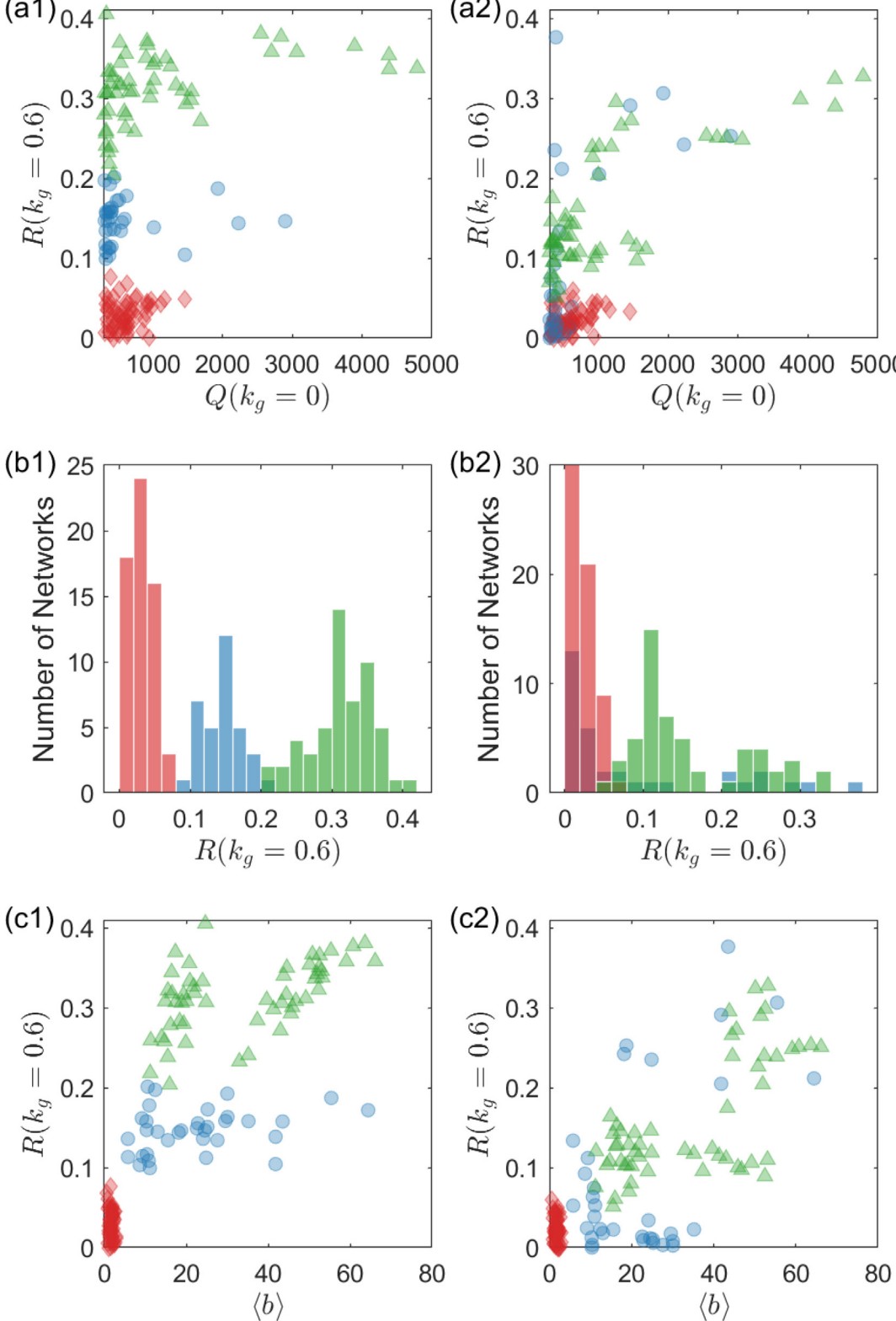

**Appendix 3—figure 3.** Dependence of the distribution of the robustness measure $R(k_g = 0.6)$ on circuit topology. (**a1**) For $J = 1$ (as in the main text), $R(k_g = 0.6)$ versus $Q(k_g = 0)$ (a quantity that measures the likelihood of a functional circuit) for all 425 network topologies. (**b1**) Histogram of $R(k_g = 0.6)$ for $J = 1$ constructed from all network topologies. (**c1**) $R(k_g = 0.6)$ versus the burden parameter. In (**a1–c1**), each data point represents a specific network topology. (**a2–c2**) The same legends as in (**a1–c1**), respectively, for $J \to \infty$. Even with the burden removed, the circuit topologies in the red group remain less stable than those in the green group.

The third issue is the effect of the network topology on the survival ratio $R(k_g)$. **Appendix 3— figure 3** presents a comparison of the dependency of $R$ on the circuit topology for $J = 1$ and $J \rightarrow \infty$. As discussed above, the difference in the burden $b$ can be a major reason for the data points in the red group to have lower $R$ values compared with those in the blue and green groups. When the term $b$ is effectively removed by setting $J \rightarrow \infty$, the difference diminishes. It can be seen from **Appendix 3—figure 3a2, b2, c2** that, in this case, the range of $R$ for the red group, in spite of the low $R$ values, overlaps with that of the blue group. However, the $R$ values associated with the red group are still distinctly smaller than those with the green group, suggesting some characteristic differences in the network motifs that define these two groups.

**Appendix 3—figure 3b1, b2** indicates a persistent feature of the distribution of the survival ratio $R(k_g = 0.6)$ for the ensemble of networks: there are three peaks regardless of whether $J$ has a small or a large value. Further, the three peaks are approximately located at the same positions for $J = 1$ and $J \rightarrow \infty$. This feature provides a criterion to determine the likelihood of a given network topology being stable or unstable under growth feedback without the need to calculate the $R(k_g)$ value for many values of the feedback strength. In particular, if the network is such that its $R(k_g = 0.6)$ value is associated with the red peak, then it is highly likely to be unstable and fail to function under growth feedback. On the contrary, if a network 'belongs' to the green peak, then the chance for it to sustain its function in a growth environment will be improved significantly.

There can be two different mechanisms for growth-induced oscillations: (1) by altering the system parameter and (2) by altering the circuit topology with the additional dynamical variable $N$ and regulations attached to it. Our results suggest the first mechanism is the major one, while the second one does not appear to play a significant role. The second mechanism only exists with a finite $J$. Thus, we compare the cases of $J = 1$ and the limit of a large $J$. As shown in **Appendix 3—figure 3**, the ratio of functional failures caused by growth-induced oscillations does not change much between the two cases. However, the oscillatory behavior is sensitive to the value of the dilution parameter. In order to have oscillations, it is necessary that the parameter be in some specific interval (**Novák and Tyson, 2008**).

## Appendix 4

### Lack of correlation between the circuit robustness and topological families

As shown in *Appendix 4—figure 1*, the network topologies belonging to the two different families (marked in different colors) are mingled together and spread all over the range of $R(k_g)$, suggesting no significant correlation between the circuit robustness and circuits family. To quantify this irrelevance, we calculate the point biserial correlation between (a) the $R(k_g)$ values of all the network topologies with $Q(k_g = 0) \leq 200$ (to lower the fluctuations) and (b) a binary variable $b_f$ which is $b_f = 0$ for the NFBL family and $b_f = 0$ for the IFFL family. The calculation involves 108 NFBL network topologies and 93 IFFL topologies. The resulting point biserial correlation is as small as 0.1. The 95% confidence interval for the true difference with respect to the two families of $R(k_g)$ is (–0.01,0.06), which is narrow around zero.

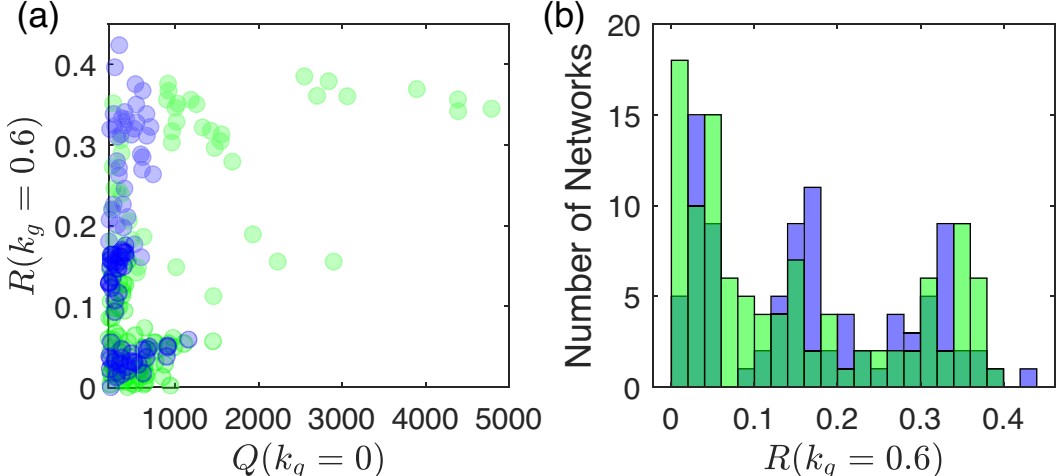

**Appendix 4—figure 1.** Demonstration of circuit robustness against growth feedback being unrelated to negative feedback loop (NFBL) or feed-forward loop (IFFL) family membership. The green and blue colors represent the NFBL and IFFL families, respectively. (**a**) Robustness measure $R(k_g = 0.6)$ versus $Q(k_g = 0)$, where each node represents a network topology. Circuits from both families are widely distributed across different levels of $R$ and intermingled. (**b**) Distributions of $R(k_g = 0.6)$ for the two families, which are quite similar.

## Appendix 5

### An analysis on the mathematical criterion for robustness against growth feedback

The quantitative measure $R(k_g)$ we have introduced to characterize the effects of growth feedback on gene circuit functioning is generally not amenable to analytic treatment. However, for weak feedback, certain analytic insights can still be gained. Here, we consider a three-node gene circuit designed to have adaptation and analyze how growth feedback destroys adaptation. We focus on type-I failure, where the growth feedback makes $O_2(C)$ deviate from $O_1(C)$, because (1) this type of failures is arguably the most important type as it alone takes nearly half of all the failures, and (2) it can be analyzed. Here, we provide a semi-quantitative analysis to elucidate how a small $k_g > 0$ can make $O_2(C)$ deviate from $O_1(C)$.

### Circuit robustness in the absence of growth feedback

The dynamical equations of the circuit in the absence of growth feedback are:

$$\frac{dA}{dt} = f_A = G_A - d_A A, \tag{16}$$

$$\frac{dB}{dt} = f_B = G_B - d_B B, \tag{17}$$

$$\frac{dC}{dt} = f_C = G_C - d_C C, \tag{18}$$

where

$$G_A = H_{\text{Input},A}(\text{Input}) \cdot H_{A,A}(A) \cdot H_{B,A}(B) \cdot H_{C,A}(C), \tag{19}$$

and each $H$ term represents the regulation of a single link in the circuit. The steady-state solutions $(A_0, B_0, C_0)$ are given by

$$A_0 = G_A/d_A, \tag{20}$$

$$B_0 = G_B/d_B, \tag{21}$$

$$C_0 = G_C/d_C. \tag{22}$$

For notation convenience, we use $x$ to denote an arbitrary node (A, B, or C). The steady-state solutions can thus be written as

$$x_0 = G_x/d_x. \tag{23}$$

With a small input signal change $\Delta I$ applied to the circuit, the steady states becomes $(A_0 + \Delta A_0, B_0 + \Delta B_0, C_0 + \Delta C_0)$. Under $\Delta I$, the dynamical equations at the steady point can be written as

$$0 = f_x(y_0 + \Delta y_0, \text{input} = \Delta I). \tag{24}$$

For $\Delta I = 0$, the equation becomes

$$0 = f_x(y_0, \text{input} = 0). \tag{25}$$

Subtracting *Equation 25* from *Equation 24*, we get

$$\begin{bmatrix} 0 \\ 0 \\ 0 \end{bmatrix} = \mathcal{J}_f \begin{bmatrix} \Delta A_0 \\ \Delta B_0 \\ \Delta C_0 \end{bmatrix} + \begin{bmatrix} \frac{\partial f_A}{\partial I} \\ 0 \\ 0 \end{bmatrix} \Delta I, \tag{26}$$

where

$$\mathcal{J}_f = \begin{bmatrix} \frac{\partial f_A}{\partial A} & \frac{\partial f_A}{\partial B} & \frac{\partial f_A}{\partial C} \\ \frac{\partial f_B}{\partial A} & \frac{\partial f_B}{\partial B} & \frac{\partial f_B}{\partial C} \\ \frac{\partial f_C}{\partial A} & \frac{\partial f_C}{\partial B} & \frac{\partial f_C}{\partial C} \end{bmatrix} \tag{27}$$

is the Jacobian matrix of the original dynamical equations evaluated at $(A_0, B_0, C_0)$.

Solving **Equation 26**, we have

$$\begin{bmatrix} \Delta A_0 \\ \Delta B_0 \\ \Delta C_0 \end{bmatrix} = -\mathcal{J}_f^{-1} \begin{bmatrix} \frac{\partial f_A}{\partial I} \\ 0 \\ 0 \end{bmatrix} \Delta I. \tag{28}$$

For the steady state to remain stable under $\Delta I$, the requirement is that ratio $\Delta C_0/\Delta I$ be small. Assuming that the Jacobian matrix satisfies the conditions to make points $(A_0, B_0, C_0)$ and $(A_0 + \Delta A_0, B_0 + \Delta B_0, C_0 + \Delta C_0)$ stable in their corresponding dynamical systems, we have

$$\frac{\Delta C_0}{\Delta I} = \left( -\mathcal{J}_f^{-1} \begin{bmatrix} \frac{\partial f_A}{\partial I} \\ 0 \\ 0 \end{bmatrix} \right)_3, \tag{29}$$

where $(\cdot)_3$ denotes the third component of the vector inside. The limiting case of a perfectly precise circuit is defined to be $\Delta C_0/\Delta I = 0$, yielding a precision criterion of by $(\mathcal{J}_f^{-1})_{31} \approx 0$ or

$$\left( \frac{\partial f_B}{\partial A} \frac{\partial f_C}{\partial B} - \frac{\partial f_B}{\partial B} \frac{\partial f_C}{\partial A} \right) / \text{Det}(\mathcal{J}_f) \approx 0. \tag{30}$$

leading to

$$\frac{\partial f_B}{\partial A} \frac{\partial f_C}{\partial B} - \frac{\partial f_B}{\partial B} \frac{\partial f_C}{\partial A} = 0. \tag{31}$$

which is the central criterion analyzed in **Shi et al., 2017**. The two families, NFBL and IFFL, satisfy this same criterion through different mechanisms.

## Precision criteria in the presence of weak growth feedback and $J \to \infty$

We now incorporate growth feedback into the analysis in the limit $J \to \infty$. In this case, the burden $b$ is small so that the dilution strength can be approximated as $dN/dt/N \approx k_g$. Suppose weak growth feedback is present before and after the small input signal $\Delta I$ is applied. Let the steady state under growth feedback before application of $\Delta I$ be denoted as $(A_0', B_0', C_0')$. The steady state with input $\Delta I$ can be written as $(A_0' + \Delta A_0', B_0' + \Delta B_0', C_0' + \Delta C_0')$. The basic equations before and after application of $\Delta I$ are

$$\frac{dx'}{dt} = f_x(y', \text{input} = 0) - k_g x', \tag{32}$$

$$\frac{d(x' + \Delta x')}{dt} = f_x(y' + \Delta y', \text{input} = \Delta I) - k_g(x' + \Delta x'). \tag{33}$$

Subtracting **Equation 32** from **Equation 33**, we get

$$\begin{bmatrix} 0 \\ 0 \\ 0 \end{bmatrix} = (\mathcal{J}_f' - k_g \mathcal{I}) \begin{bmatrix} \Delta A' \\ \Delta B' \\ \Delta C' \end{bmatrix} + \begin{bmatrix} \frac{\partial f_A}{\partial I} \\ 0 \\ 0 \end{bmatrix} \Delta I, \tag{34}$$

where $\mathcal{I}$ is the identity matrix. The solution is

$$\begin{bmatrix} \Delta A' \\ \Delta B' \\ \Delta C' \end{bmatrix} = -(\mathcal{J}_f' - k_g \mathcal{I})^{-1} \begin{bmatrix} \frac{\partial f_A}{\partial I} \\ 0 \\ 0 \end{bmatrix} \Delta I. \tag{35}$$

Compared with *Equation 28*, the differences are that the matrix $\mathcal{G}_f$ is replaced by $(\mathcal{J}_f' - k_g \mathcal{I})$, and $x, \Delta x$ are replaced by $x', \Delta x'$, respectively.

The precision criterion again requires $\Delta C_0'/\Delta I$ to be small. We have

$$\frac{\Delta C_0'}{\Delta I} = \left( -(\mathcal{J}_f' - k_g \mathcal{I})^{-1} \begin{bmatrix} \frac{\partial f_A}{\partial I} \\ 0 \\ 0 \end{bmatrix} \right)_3, \tag{36}$$

which is equivalent to

$$((\mathcal{J}_f' - gE)^{-1})_{31} = \left[ \frac{\partial f_B}{\partial A} \frac{\partial f_C}{\partial B} - (\frac{\partial f_B}{\partial B} - k_g)\frac{\partial f_C}{\partial A} \right]_{A',B',C'} / \text{Det}(\mathcal{J}_f' - g\mathcal{I}) \approx 0, \tag{37}$$

leading to

$$\left[ \frac{\partial f_B}{\partial A} \frac{\partial f_C}{\partial B} - (\frac{\partial f_B}{\partial B} - k_g)\frac{\partial f_C}{\partial A} \right]_{A',B',C'} \approx 0. \tag{38}$$

Comparing this equation for precision criterion *Equation 38* with the criterion *Equation 31* in the absence of growth feedback, we find an extra term of $k_g$. This explicit term of $k_g$ makes the criterion more difficult to satisfy with a range of different $k_g$ values. It requires either $\partial f_C/\partial A$ is zero or the four partial derivative terms change accordingly with a varying $k_g$ to have exact cancellations.

For neither the NFBL nor the IFFL family, $\partial f_C/\partial A = 0$ can be satisfied. In none of the 425 network topologies, the link from node A to C is absent ($\partial f_C/\partial A = 0$). Thus with a random sampling of the parameters for the circuits that have adaptation at $k_g = 0$, the probability that $\partial f_C/\partial A = 0$ can occur is negligibly small.

## Precision criterion with exact cancellations for the optimal family

As the criterion $\partial f_C/\partial A = 0$ cannot be satisfied in three-node gene circuits, we discuss the possibility of exact cancellations with varying $k_g$. For the optimal circuit family demonstrated in *Figure 5b*, we have $\partial f_C/\partial B = 0$ as there is no direct link from node B to C. The precision criterion becomes

$$\left[ (\frac{\partial f_B}{\partial B} - k_g)\frac{\partial f_C}{\partial A} \right]_{A_0',B_0',C_0'} \approx 0. \tag{39}$$

Since $\partial f_C/\partial A \neq 0$, this can be rewritten as

$$\frac{\partial f_B}{\partial B}|_{A_0',B_0',C_0'} - k_g \approx 0. \tag{40}$$

For this family, the precision criterion in the absence of growth feedback is

$$\frac{\partial f_B}{\partial B}|_{A_0,B_0,C_0} \approx 0. \tag{41}$$

Combining *Equations 40 and 41*, we get

$$\frac{\partial^2 f_B}{\partial A \partial B}|_{A_0,B_0,C_0}(A_0' - A_0) + \frac{\partial^2 f_B}{\partial C \partial B}|_{A_0,B_0,C_0}(C_0' - C_0)$$

$$\approx k_g. \tag{42}$$

Using the approximation employed in *Shi et al., 2017* for the NFB family that $f_B$ is a linear function of $B$, we have

$$\frac{\partial f_B(A, B, C)}{\partial B} \approx \frac{f_B(A, B, C)}{B}$$

$$= \frac{v_B}{K_{BB}} H_{A,B}(A) H_{C,B}(C) - d_B. \tag{43}$$

We thus have

$$\frac{dH_{A,B}(A)}{dA}|_{A_0} H_{C,B}(C_0)(A'_0 - A_0) + H_{A,B}(A_0)\frac{dH_{C,B}(C)}{dC}|_{C_0}(C'_0 - C_0) \approx \frac{K_{BB}}{v_B} k_g. \tag{44}$$

This equation can be solved analytically only in the regime of $k_g \sim 0$ where $(A'_0 - A_0)$ and $(C'_0 - C_0)$ are approximately linear functions of $k_g$. But it should be difficult for the circuit to meet this criterion with a random sampling of the circuits that have adaptation at $k_g = 0$.

## Appendix 6

### Network motifs supporting oscillations

As summarized in *Novák and Tyson, 2008*, three classes of motifs can support oscillations in a three-node circuit.

### Class 1 (the dominant class)

Delayed negative-feedback loop with an intermediate node in the path of the NFBL. A majority of the networks with an oscillation-supporting motif belong to this class (237 out of 245 networks). All the circuits that have more than 20% failures as oscillation-induced failures belong to this class.

### Class 2

Amplified negative-feedback loop, with a node regulated by both a negative-feedback loop through another node and a positive-feedback loop through the third node. There are only eight network topologies that fall into this class. They result in 3–20% oscillation-induced failures.

### Class 3

Incoherently amplified negative-feedback loops, as demonstrated in Figure 5c of *Novák and Tyson, 2008*. Among all the 425 networks capable of adaptation studied in our work, no network belongs to this class.

## Appendix 7

### Oscillation-related bifurcations

In growth-feedback-induced oscillations, we mentioned the oscillation-related bifurcations that can give birth to undesired oscillation to the circuit and cause adaption failure. Here, we provide further details on the two types of bifurcations: saddle-node bifurcation of cycles and infinite-period bifurcation. In this appendix, we use $t_{\text{step}} = 0.01$ and $t_{\text{block}} = 1,000$ to make the results more accurate.

In *Appendix 7—figure 1*, we show a demonstration of an infinite-period bifurcation. In panel (b), there is the characteristic slow–fast behavior before such a bifurcation. On the limit cycle, there are ranges where the system rotates fast (with a sharp slope in our panel) or slow (with an almost flat curve in our panel). These 'slow parts' are slow passage through a bottleneck due to a ghost point that is close to becoming a fixed point (*Strogatz, 2018*). From panel (e) to panel (a), as $k_g$ decreases, the 'slow parts' in the curves become slower and slower, until the period length diverges where the ghost point actually becomes a fixed point. This trend of period length is also shown in panel (f), where the period length diverges at the critical $k_g$.

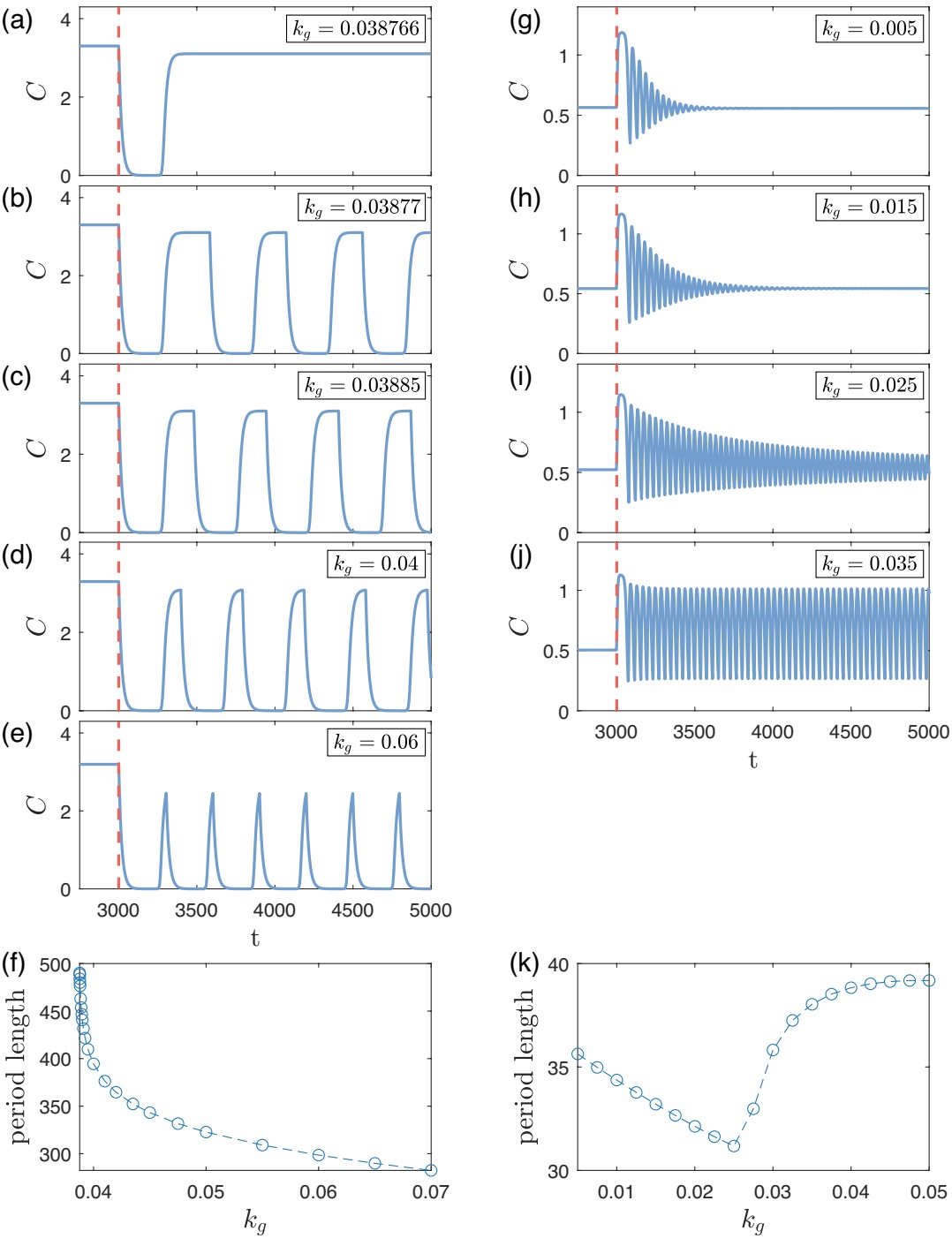

**Appendix 7—figure 1.** Demonstration of oscillation-related bifurcations. Here we demonstrate two types of oscillation-related bifurcations: (**a–f**) infinite-period bifurcation and (**g–k**) saddle-node bifurcation of cycles. Panels (**a–e**) show the output signal of a circuit around an infinite-period bifurcation with increasing $k_g$. The circuit fails between panels (**a**) and (**b**) due to the emergent oscillation. Panel (**f**) shows the length of the oscillation period with respect to $k_g$ after the critical $k_g$ value. There is no persisting oscillation before the critical $k_g$. Panels (**g–j**) show the output signal of another circuit around a saddle-node bifurcation of cycles with increasing $k_g$. The circuit failed between panels (**i**) and (**j**). Panel (**k**) shows the length of the oscillation period with respect to $k_g$ after the critical $k_g$ value, where two distinct branches exist on the two sides of the critical $k_g$.

In *Appendix 7—figure 1g–k*, we show a demonstration of a saddle-node bifurcation of cycles. When $k_g$ is small, there is only a damped oscillation that quickly approaches a fixed point. As $k_g$ increases from panel (g) to panel (j), the damping is weaker and weaker until the outer cycle becomes

stable. In panel (k), we show the period length with respect to various $k_g$ values. More precisely, here we are plotting the limits of the period length, as the period length can have some transient values before approaching the limit. The period length before and after the critical $k_g$ (which is somewhere between 0.025 and 0.0275) has two distinct branches. Although these two branches have different trends, they have relatively similar values around the critical $k_g$ point, without the divergence behavior around an infinite-period bifurcation. This is one of the features of a saddle-node bifurcation of cycles.

## Appendix 8

### Regularized feed-forward neural networks for identifying critical links

We employed ensembles of regularized feed-forward neural networks to detect, in an automated fashion, the links that are crucial in determining the level of robustness $R$. The neural-network structure is illustrated in *Figure 6e*, which has three layers: an input layer, a hidden layer, and an output layer. The input layer receives a nine-dimensional circuit topology vector where each entry represents a potential link in the three-node circuit, such as $A \rightarrow A$ and $B \rightarrow C$. For an activation (inhibition) link, the entry value is set to +1 (−1). In the absence of such a link, the value is zero. In the hidden layer, there are only two neurons that use a hyperbolic tangent activation function, creating a bottleneck that limits the complexity of the extracted features. The output layer has one neuron that uses a hyperbolic tangent activation function trained to output the estimated robustness $\hat{R}$. The input and hidden layers are connected by the matrix $W_{\text{in}}$, and the hidden and output layers are connected by the matrix $W_{\text{out}}$. Given the input vector $u$, the estimated $\hat{R}$ can be expressed as

$$\hat{R} = \tanh[W_{\text{out}} \tanh(W_{\text{in}} u)]. \tag{45}$$

We use all the 303 circuit topologies that have $Q(k_g = 0) > 100$ for training to minimize the relative random fluctuations in the training data. The loss function for optimization is

$$\text{Loss} = |\hat{R} - R| + \beta \sum_{i=1}^{w_h} \sum_{j=1}^{L_n} |W_{\text{in},ij}|, \tag{46}$$

where $\beta = 0.05$ is the $l - 1$ regularization coefficient, $L_n = 9$ is the number of possible links within a three-gene circuit, and $w_h = 2$ is the width of the hidden layer. We train the network using a stochastic gradient descent algorithm and repeat it 50 times with different initial weights in the neural net matrices. The 'importance' of a link is determined by the logarithm of the absolute value of the weights in $W_{\text{in}}$ corresponding to the gain of that link. This importance measure is then averaged over all 50 neural networks.

