## [Editor Report · eLife Assessment]

The paper presents **valuable** computational findings on how growth feedback affects the performance of synthetic gene circuits designed for adaptive responses. By systematically analyzing over four hundred circuit topologies, the authors provide **solid** evidence for their conclusions on failure mechanisms and design features that enhance robustness against growth dynamics. While the study's significance and rigor are somewhat constrained by its reliance on previously published network topologies, these results are highly relevant for advancing the engineering of gene circuits in various applications.

---

## [Referee Report · Joint Public Review]

Engineered artificial gene regulatory networks ("circuits") have a wide range of applications, but their design is often hindered by unforeseen interactions between the host and circuit processes. This manuscript employs computational modeling to investigate how growth feedback influences the performance of synthetic gene circuits capable of adaptation. By analyzing 425 hypothetical circuits previously identified as achieving nearly perfect adaptation (Ma et al., 2009; Shi et al., 2017), the authors introduce growth feedback into their models using additional terms in ordinary differential equations. Their simulations reveal that growth feedback can disrupt adaptation dynamics in diverse ways but also identify core motifs that ensure robust performance under such conditions. Additionally, they establish a scaling law linking circuit robustness to the strength of growth feedback. The findings have important implications for synthetic biology, where host-circuit interactions frequently compromise desired behaviors, and for systems biology, by advancing the understanding of network motif dynamics. The authors' classification schemes will be highly valuable to the community, offering a framework for addressing growth-related challenges in circuit design.

Strengths

- A detailed investigation into the reasons for adaptation failure upon the introduction of cell growth was conducted, distinguishing this work from other studies of functional screening in gene regulatory network topologies. The comprehensiveness of the analysis is particularly noteworthy.

- Approaches for assessing robustness, such as the survival ratio Q, were employed, providing tools that may be applicable to a broad range of network topologies beyond adaptation. The scaling law derived from these approaches is both novel and insightful.

- A thorough numerical analysis of three gene regulatory networks exhibiting adaptation was performed. For each of the 425 topologies analyzed, approximately 2e5 circuits were sampled using Latin hypercube sampling, ensuring robust coverage of the parameter space. Among these, 1.5e5 circuits were identified as showing adaptation and subsequently subjected to further analysis, yielding approximately 350 parametric designs per topology for deeper investigation.

- The systematic approach and depth of the analysis position this study as a significant contribution to the understanding of gene regulatory networks and their response to growth feedback. The combination of detailed investigation, novel robustness metrics, and rigorous computational techniques enhances the impact of this work within the field.

Weaknesses

- The study focuses exclusively on a preselected set of 425 topologies previously shown to achieve adaptation, limiting the exploration of whether growth feedback could enable adaptation in circuits not inherently adaptive. While the authors have discussed and justified this choice, the focus restricts the generality of the conclusions, as the potential for growth feedback to induce adaptation in non-adaptive circuits remains unaddressed. The analysis includes scenarios where higher growth feedback restores adaptation in circuits that lose it at intermediate levels, but further elaboration on the implications for circuit design would strengthen the impact. The numerical framework and parameter choices align well with established methods, and an overview of the selected topologies has been provided. However, offering detailed information in supplementary materials or a public repository would further enhance the paper's accessibility and reproducibility.

- The model fails to capture the influence of protein levels on growth. To ensure accurate modeling of protein-level effects on growth, the b(t) term should be scaled appropriately, similar to Tan et al. Nature Chemical Biology 5:842-848 (2009).

- The authors propose bistability or multistability as the primary mechanisms behind different types of adaptation failure, explaining why the failures do not occur precisely at bifurcation points. They argue that their ODE simulations provide evidence for oscillation-related bifurcations, and an included appendix explores this phenomenon further, detailing how it can be observed in their results. While the authors choose not to apply semi-analytic methods, such as numerical continuation and eigenvalue analysis, to validate the existence of bifurcations, their approach offers valuable insights into the underlying dynamics of adaptation failures.

- The analysis in this work is carried out exclusively in a deterministic regime, as the focus is on scenarios where the effects of noise are assumed to be minimal. This approach is justified, and the authors acknowledge the complexity of extending their analysis to include stochasticity, which they suggest as an avenue for future research. The discussion has been expanded to address the potential impact of noise, its handling, and the assumptions underlying its exclusion. It is important to note, however, that noise can significantly alter system behavior-for instance, stabilizing trajectories and removing oscillations, as shown in prior studies (e.g., 10.1016/j.cels.2016.01.004). Additionally, variability in experimental implementations may influence the dynamics beyond what is predicted in deterministic models. These factors should be considered when interpreting the results.

---

## [Author Response]

The following is the authors’ response to the original reviews.

**Point-by-point response to the public review:**
General Comment: “Using computational modeling, this manuscript explores the effect of growth feedback on the performance of gene networks capable of adaptation. The authors selected 425 hypothetical synthetic circuits that were shown to achieve nearly perfect adaptation in two earlier computational studies (see Ma et al. 2009, and Shi et al. 2017). They examined the effects of cell growth feedback by introducing additional terms to the ordinary differential equation-based models, and performed numerical simulations to check the retainment and the loss of the adaptation responses of the circuits in the presence of growth feedback. The authors show that growth feedback can disrupt the gene network adaptation dynamics in different ways, and report some exceptional core motifs which allow for robust performance in the presence of growth feedback. They also used a metric to establish a scaling law between a circuit robustness measure and the strength of growth feedback. These results have important implications in the field of synthetic biology, where unforeseen interactions between designed gene circuits and the host often disrupt the desired behavior. The paper’s conclusions are supported by their simulation results, although these are presented in their summary formats and it would be useful for the community if the detailed results for each topology were available as a supplementary file or through the authors’ GitHub repository.”

We are grateful for the referee’s positive evaluation of our work. We have updated our GitHub and OSF repositories with detailed results for each topology. Additionally, we have included other simulation codes, result data, and detailed explanations in these two repositories that may be of interest to our readers.

Strength 1: “This work included a detailed investigation of the reasons for adaptation failure upon introducing cell growth to the systems. The comprehensiveness of the analysis makes the work stand out among studies of functional screening of network topologies of gene regulation.”

We are grateful for the referee’s positive assessment of our work, notably the recognition of the ‘detailed investigation’ we conducted, and the ‘comprehensiveness of the analysis’ we provided.

Strength 2: “The authors’ approaches for assessment of robustness, such as the survival ratio Q, can be useful for a wide range of topologies beyond adaptation. The scaling law obtained with those approaches is interesting.”

We are grateful for the referee’s positive evaluation of our defined factors for assessing circuit robustness. We also appreciate the acknowledgment of the “interesting” nature of the scaling law we discovered using the assessment factor *R*.

Weaknesses 1: “The title suggests that the work investigates the ’effects of growth feedback on gene circuits’. However, the performance of ’nearly perfect adaptation’ was chosen for the majority of the work, leaving the question of whether the authors’ conclusion regarding the effects of growth feedback is applicable to other functional networks.”

We agree that our present title can be too broad, and we have changed it from “Effects of growth feedback on gene circuits: A dynamical understanding” to “Effects of growth feedback on *adaptive* gene circuits: A dynamical understanding”. Although we have some brief results and discussions on the gene circuits with bistability, we admit that most of our results and discussions are focused on circuits that have adaptation.

The new title is more specific and should be a more appropriate summary of the paper.

Weaknesses 2: “This work relies extensively on an earlier study, evaluating only a selected set of 425 topologies that were shown to give adaptive responses (Shi et al., 2017). This limited selection has two potential issues. First, as the authors mentioned in the introduction, growth feedback can also induce emerging dynamics even without existing function-enabling gene circuits, as an example of the ”effects of growth feedback on gene circuits”. Limiting the investigation to only successful circuits for adaptation makes it unclear whether growth feedback can turn the circuits that failed to produce adaptation by themselves into adaptation-enabling circuits. Secondly, as the Shi et al. (2017) study also used numerical experiments to achieve their conclusions about successful topologies, it is unclear whether the numerical experiments in the present study are compatible with the earlier work regarding the choice of equation forms and ranges of parameter values. The authors also assumed that all readers have sufficient understanding of the 425 topologies and their derivation before reading this paper.”

We agree with the reviewer that several issues need to be clarified in our new manuscript. We have added new discussions for all of them.

We agree with the reviewer that growth feedback could turn the non-adaptive circuits into adaptationenabling circuits, and this indeed presents a compelling topic for future research. We have added the following discussions to our paper, talking about a relevant matter. We find that in our simulated dataset, there are cases where a higher degree of growth feedback can restore the adaptation that has been lost in a circuit. However, as we discussed in this new paragraph, a comprehensive study in the direction of turning non-adaptive circuits into adaptation-enabling circuits will “require entirely different approaches for sampling circuit parameters and selecting candidate network topologies, demanding significantly high computational costs.” Given that this topic extends beyond the scope of the current paper, we leave this matter to future research.

“Although the primary focus of this paper is on how growth feedback can undermine an originally adaptive circuit and how to design circuits that are robust against such feedback, our simulated dataset reveals instances where growth feedback can benefit the circuit within certain ranges. Specifically, we identified 2,092 circuits across 306 different topologies where adaption, lost at an intermediate level of growth feedback, is restored at higher levels. This is 1.4% of all circuits tested. We anticipate that additional circuits exhibiting this loss-and-recovery behavior exist, as our sampling of six discrete levels of *kg* (0,0.2,0.4,0.6,0.8,1.0) might have overlooked numerous cases. This result again suggests the possible advantages of growth feedback in gene circuits (Tan et al., 2009; Nevozhay et al., 2012; Deris et al., 2013; Feng et al., 2014; Melendez-Alvarez and Tian, 2022). A comprehensive study into how growth feedback can endow or enhance adaption in circuits would require entirely different approaches for sampling circuit parameters and selecting candidate network topologies, demanding significantly high computational costs. Given that this topic extends beyond the scope of the current paper, we leave this matter to future research.”

We have added the following discussions about the reasoning behind using the 425 network topologies selected from the study Shi et al. (2017).

“We use these 425 network topologies from the study (Shi et al., 2017), avoiding redundancy with established results. Due to the unique focus of our research on the effects of growth feedback and the need to evaluate quantitative ratios of robust circuits among all functional ones, we have chosen to use a 20-fold increase in the number of random parameter sets for each network topology compared to the simulations in (Shi et al., 2017). This approach makes it computationally prohibitive to scan all possible 16,038 three-node circuits. We carefully follow the settings in (Shi et al., 2017), which also analyzed TRNs with the AND logic as in this paper. Detailed descriptions of our simulation experiments are provided in the Methods section. To make our results more convincing, we have adopted a set of adaptation criteria that are stricter than those used in (Shi et al., 2017). Consequently, the ratio of adaptive circuits is somewhat lower in our study, with 4 out of the 425 network topologies not demonstrating adaptation.”

Other than the more strict adaptation criteria and much larger sampling sizes, as we mentioned in this paragraph, we have carefully followed the simulation details of the study Shi et al. (2017). This includes but is not limited to: the dynamical equations (when *kg* = 0), the input signals, the scales and ranges of the circuit parameters to be randomly sampled, and the sampling method (Latin hypercube sampling). One of the authors of the current paper was also the first author of the study Shi et al. (2017), who helped us verify the details of simulations (among many other contributions). These identical settings justify our usage of the established results with the 425 network topologies.

To provide more information about these 425 network topologies, We have added the following introduction. It introduces the structural features of the networks, especially the shared core motifs for adaptation. In our GitHub and OSF repositories, we have also provided relevant data about the 425 topologies, including the topology structures and the parameter sets we scanned.

“These topologies can be classified into two families based on the core topology: networks with a negative feedback loop (NFBL) and networks with an incoherent feed-forward loop (IFFL) (Shi et al., 2017). More specifically, there are 206 network topologies in the NFBL family. All of these NFBL topologies have a negative feedback loop for node B. This negative feedback loop can be formed by the loop from node B to A and back to B (such as the circuit shown in Fig. 1 (a)), by node B to C and back to B, or by a longer route, from node B to A and then to C and back to B. There is always a self-activation link from B to B in all these 206 NFBL networks. There are 219 network topologies in the IFFL family. All of them have two feed-forward pathways from the input node A to the output node C. One pathway goes from node A to C directly, while the other involves node B in the middle. One of the pathways is activating while the other one is inhibitory.”

Weaknesses 3: “The authors’ model does not describe the impact of growth via a biological mechanism: they model growth as an additional dilution rate and calculate growth rate based on a phenomenological description with growth rate occurring at a maximum (*kg*) scaled by the circuit ’burden’ b(t). Therefore, the authors’ model does not capture potential growth rate changes in parameter values (e.g., synthetic protein production falls with increasing growth rate; see Scott & Hwa, 2023).”

In our paper, we consider dilution due to cell growth as the dominant factor of growth feedback. Here we compared the adaptive circuits under no-growth conditions and their ability to maintain their adaptive behaviors after dilution into a fresh medium, which mediated a significant dilution to the circuits. This is based on our previous work, Zhang, et al. Nature chemical biology 16.6 (2020): 695-701. We agree that an increased growth rate can change synthetic protein production. However, the dynamic roles of the dilution and growthaffected production rate should be analogous, given that they both act as inhibitory factors arising from cell growth as mentioned by the reviewer. Still, we agree that taking the growth effect on the production rate into account would provide a more comprehensive study, but it is beyond the scope of the present work. We have added the following paragraph in the Discussion section of our paper.

“In our paper, we consider dilution due to cell growth as the dominant factor of growth feedback. Here we compared the adaptive circuits under no-growth conditions and their ability to maintain their adaptive behaviors after dilution into a fresh medium, which mediated a significant dilution to the circuits. This is based on our previous work (Zhang et al. (2020)). However, growth feedback is inherently complex (Klumpp et al. (2009)). For instance, an increased growth rate can change protein synthesis rate (Hintsche and Klumpp (2013); Scott and Hwa (2023)), and cell growth rates can affect the distribution of protein expression in cell populations (Gouda et al. (2019)). In our paper, we concentrate on a simplified model with dilution, which we consider to have captured the dominant factor. The dynamic roles of the dilution and growth-affected production rate should be analogous, given that they both act as inhibitory factors arising from cell growth. Incorporating the impact of growth rate on protein synthesis into our model would offer a more comprehensive analysis, a task beyond the scope of this paper but presenting an intriguing opportunity for future research to address the complexities of growth feedback.”

Weaknesses 4: “The authors made several claims about the bifurcations (infinite-period, saddle-node, etc) underlying the abrupt changes leading to failures of adaptations. There is a lack of evidence supporting these claims. Both local and global bifurcations can be demonstrated with semi-analytic approaches such as numerical continuation along with investigations of eigenvalues of the Jacobian matrix. The claims based on ODE solutions alone are not sound.”

After our further simulations and verification, we found that most of the bifurcation-induced failures we mentioned in type-V and type-VI failures should be categorized as bistability or multistability-induced failures. They are still abrupt switching between adaptive and non-adaptive states, as we described in the previous version of the manuscript. However, they are actually still far away from the bifurcation points at the critical *kg*. We have corrected all relevant descriptions and figures, including panel Fig. 4 (c) and its captions. We have added the following paragraph in the paper to explain this issue.

“One might expect bifurcations to play an important role in many type-V and type-VI failures. However, in our simulations, failures precisely at the bifurcation point are not observed. This is because the bifurcation points under consideration, such as fold bifurcations, are where one of the attraction basins diminishes to zero. For a failure to occur exactly at the bifurcation point, the initial condition would need to coincide precisely with the infinitesimally small basin just before it vanishes. More realistically, failures almost always largely precede the exact bifurcation point. They happen while the basin is still contracting and the basin boundary crosses the initial condition or *O*_1_. An example is shown in Fig. 4(b), where bistability persists, yet the lighter orange basin with a larger *O*_1_(C) cannot be reached as the boundary shifts away from the initial condition *A*_0_ and *B*_0_. As another example, in Fig. 4 (c) from a different circuit, the higher *O*_2_(*C*) state disappears at *kg* ≈ 0.012 and switches to a lower *O*_2_(*C*), but this point is not a bifurcation.

It is the point where the stable *O*_1_ continuously crosses the basin boundary of *O*_2_.”

Our further simulations have verified the existence of the oscillation-related bifurcations. We have added a new appendix discussing the phenomena associated with them in more detail.

Weaknesses 5: “The impact of biochemical noise is not evaluated in this work; the author’s analysis is only carried out in a deterministic regime.”

In this paper, we have not taken into account biochemical noise as we focus solely on scenarios where all protein concentrations are high. In these circumstances, the influence of noise is relatively minor. Incorporating biochemical noise, which originates from various sources and possesses diverse characteristics, would significantly complicate the analysis beyond the scope of our current work. However, exploring this aspect could be an intriguing avenue for future research. We have included the following discussions in our paper.

“Our study focuses on scenarios where random noises are ignored. Realistically, gene circuits are subjected to diverse types of noise, which can complicate their predictable behavior and design. These noises can originate externally from a noisy input signal I, or intrinsically, directly affecting the circuit components. Further, these noises can be classified based on various mechanisms that cause them (Colin et al. 2017; Sartori and Tu 2011) . And with different mechanisms, each type of noise can be characterized by different attributes such as frequency, amplitude, and noise color. These variances can lead to different impacts on the circuits, potentially necessitating unique mechanisms or designs for the attenuation of each category (Sartori and Tu 2011; Qiao et al. 2019). Given the extensive complexity and the need for thorough investigation, these noise-related challenges are beyond the scope of this paper and require a series of future studies.”

**Point-by-point response to the recommendations for the authors:**
Comment 1: - The authors’ github repository, detailed in their code availability statement, is currently unavailable and likely contains some of the answers to the queries here.

We have updated our GitHub and OSF repositories with simulation codes, result data, and detailed explanations. The link to our GitHub repository in the previous version of the manuscript contained a format error, making it inaccessible to the referees. We apologize for this mistake and have corrected it.

Comment 2: - At present, it is not clear how the 425 topologies are created from the system of equations (Eq. 6-8) or from the circuit diagram in Fig 1a. This could do with being explicitly stated for the reader.

We have added the following paragraph to discuss how the 425 topologies are selected and what the common motifs and connections they share.

“Previous research identified 425 different three-node TRN network topologies that can achieve adaptation in the absence of growth feedback (Shi et al., 2017), providing the base of our computational study. These topologies can be classified into two families based on the core topology: networks with a negative feedback loop (NFBL) and networks with an incoherent feed-forward loop (IFFL) (Shi et al., 2017). More specifically, there are 206 network topologies in the NFBL family. All of these NFBL topologies have a negative feedback loop for node B. This negative feedback loop can be formed by the loop from node B to A and back to B (such as the circuit shown in Fig. 1 (a)), by node B to C and back to B, or by a longer route, from node B to A and then to C and back to B. There is always a self-activation link from B to B in all these 206 NFBL networks. There are 219 network topologies in the IFFL family. All of them have two feed-forward pathways from the input node A to the output node C. One pathway goes from node A to C directly, while the other involves node B in the middle. One of the pathways is activating while the other one is inhibitory. We use these 425 network topologies from the study (Shi et al., 2017), avoiding redundancy with established results. Due to the unique focus of our research on the effects of growth feedback and the need to evaluate quantitative ratios of robust circuits among all functional ones, we have chosen to use a 20-fold increase in the number of random parameter sets for each network topology compared to the simulations in (Shi et al., 2017). This approach makes it computationally prohibitive to scan all possible 16,038 three-node circuits. We carefully follow the settings in (Shi et al., 2017), which also analyzed TRNs with the AND logic as in this paper. Detailed descriptions of our simulation experiments are provided in the Methods section. To make our results more convincing, we have adopted a set of adaptation criteria that are stricter than those used in (Shi et al., 2017). Consequently, the ratio of adaptive circuits is somewhat lower in our study, with 4 out of the 425 network topologies not demonstrating adaptation.”

Comment 3: - In the main text, the authors mentioned that they chose 425 network topologies for this study, whereas the number is 435 in the abstract. Please correct the error.

The number 435 in our previous abstract referred to the 10 four-node circuits that we studied in the appendix, in addition to the 425 three-node network topologies. To avoid confusion and potential misunderstandings among readers, we have revised this expression of “435 distinct topological structures” to “more than four hundred topological structures”.

Comment 4: - Please can the authors include the topologies they have studied in an appendix or as supplementary material. The impact of this work would increase significantly if for each topology the authors could include a pie chart similar to the one shown in Fig 2 so that others can use these results.

We fully acknowledge the potential benefits of providing simulation results for each topology. However, including over four hundred more figures in this paper is not feasible. Moreover, we expect that many readers may also be interested in results not only for individual topologies but also for subsets sharing specific motifs or regulatory connections. Therefore, we have provided all the necessary data and codes in our GitHub repository to make these pie charts. We have included a detailed guide on how to generate these pie charts in the GitHub Readme file. These allow readers to plot the pie chart and extract distributions for any individual topology or use conditions to filter any subset of topologies as required. We believe this approach offers greater flexibility for our readers. We have also added the following explanation in the Methods section.

“The codes implementing these criteria are available in our GitHub repository, with the link provided in the ”Code Availability” section. The failure type results for all circuits tested are available in our OSF repository, with the link provided in the ”Data Availability” section. An additional note is provided in the README file of our GitHub repository for further guidance on generating pie charts similar to Fig. 2 for any network topology or subset of topologies.”

Comment 5: - At present, the authors have not given sufficient detail for their numerical methods (e.g. to identify bistability or oscillations) to enable the work to be repeated. I would appreciate it if the authors could expand their Methods section or provide a description of their method as an appendix. Additionally, the authors must clarify how many parameter sets per topology showed successful adaptation.

In response to this comment, we have reorganized and expanded our Methods section, especially the new “Numerical simulations of circuit dynamics” and “Numerical criteria for functional adaptation and failure types” subsections. We added details on how we define and evaluate a “relatively steady state”, how to determine if there is an oscillation, how to determine the critical *kg* value, and how to determine if a failure is continuous or abrupt. Readers can also find the corresponding codes in our GitHub repository, where we provide a README file to help the readers locate the script file they need.

The number of parameter sets per topology showed successful adaptation is precisely our definition of the Q-value. Q-values of most of the circuits we tested are shown in multiple figures in the paper. A complete table of Q-values with different topologies and different *kgrowth* values can be found in our OSF repository.

Comment 6: - Looking at the Model Description, there seem to be multiple issues, as follows. The model should be rewritten and all simulations redone with the model corrected as described below:(a) The ”strength of growth feedback” is modeled by the maximal growth parameter *kg* in Equation (12). However, this rate does not represent growth feedback. In fact, this parameter must be present also for the system without growth feedback, Equations (6 - 8), because those cells grow as well! So Equation (12) with b(t)=0 should also be added to Equations (6 - 8), in addition to the dilution terms in each equation.(b) The dilution due to growth (dN/dt)*(B/N) is only added to Equations (9 - 11). This is wrong - growthaffects (dilutes) all protein concentrations, even without growth feedback, so similar terms must be added even to equations without growth feedback, i.e., to Equations (6 - 8).(c) The term representing growth feedback is actually the fraction 1/(1+b(t)). To adjust the strength ofgrowth feedback, some parameters should be introduced into this term. Specifically, the term currently has a Hill form with Hill coefficient = 1 and sensitivity = 1. The term should be converted into a general Hill function, and the parameters of that function should be altered to represent growth feedback. This Hill function is called a cellular (phenotypic) fitness landscape, see Nevozhay et al., 2012.

Equations (6-8) only describe one part of the entire model we are studying. We are having these equations presented solely for the purpose of not overwhelming readers with a large number of parameters that are defined for the first time. They are not actually used in our simulations, but were only for explanations of the meaning of parameters. In our simulations throughout the paper, we only used Eqs. (9-13) (with various topologies). We have revised the texts to make this point clear. We have added the following descriptions in the section Model Description:

“In order not to overwhelm readers with too many terms and parameters, we first describe a partial model (an isolated circuit without growth feedback) before introducing the complete model that we study in this work.”

“Equations. (9) to (13) are the dynamical equations we actually use for simulating the circuit dynamics.”

Additionaly, in the newly added subsection “Numerical simulations of circuit dynamics683” in the Methods, we explicitly mention that:

“The dynamical equations we use are similar to Eqs. (9-13) but with different topologies.”

We consider dilution due to cell growth as the dominant factor of growth feedback. In fact, we study the adaptive circuits without growth and their ability to maintain their adaptive behaviors after dilution into a fresh medium, based on a recent work [Zhang, et al., Nature Chemical Biology 16.6 (2020): 695-701]. The dynamic roles of the dilution and growth-affected production rate should be analogous, given that they both act as inhibitory factors arising from cell growth. The term mentioned in the comment is about how the burden of the circuit affects cell growth. We agree that it can be interesting to have a more comprehensive study on how different degrees of nonlinearity of this term can have different effects on the overall robustness towards the growth feedback problem, but this is not part of our primary focus and is beyond the scope of this paper. In this paper, we are mostly concerned with the variability of the strength of the growth feedback/dilution, controlled by the parameter *kg*, instead of the different types of nonlinearity.

Comment 7: - On the right side of Equation (7), the first term should be inhibitory, right?

This is indeed an error. We accidentally reversed the regulation from A to B and B to A when inputting the formula. We have corrected both terms.

Comment 8: - It seems to me that a better transition from Figs 6 and 7 to Fig 8 can be made. Did the authors choose the three circuits in Fig 8 based on the three distinct groups shown in Fig 6 and 7? The rationale for choosing the three topologies given the clusters identified earlier can be explained more clearly.

We agree more explanation can be provided here. We have added the following descriptions, in the caption of Fig.8:

“The other three curves represent circuits with different robustness levels: high (Circuit No. 98), moderate (Circuit No. 3), and low (Circuit No. 28) values of *R*, to demonstrate that this scaling behavior is generic. Each of these three circuit topologies is selected from one of the three groups illustrated in Fig. 6 and Fig. 7, and they have the highest *Q*(*kg* = 0) value within their respective groups.”

and in the main text:

“The three other curves represent circuit topologies that have a relatively high, moderate, and low value *R* among the 425 topologies tested, to demonstrate that this scaling behavior is generic. These three topologies are the highest *Q*(*kg* = 0) topology in each of the three groups shown in Fig. 6 and Fig. 7.”

Comment 9: - The insights from the neural network model seem to be very limited. It would be interesting to see if the model can predict the performance of network topologies that have not been exposed to the model during training.

Machine learning is not a focus of this paper. For the section the comment was referring to, the main research question is on the relationship between circuit robustness and topology, and the point we are trying to make is that the robustness dependency varies across different connections — some connections are critical, while others are less impactful. The neural-network-based analysis was only used to provide further support to this point by demonstrating that through optimization, neural networks automatically assign different levels of weights to different connections in the circuits.

We agree that it can be an interesting topic to study how machine learning can be used to help us design functional and robust circuits, as discussed in the final paragraph of the Discussion section. However, such an investigation would require a series of more comprehensive and carefully designed simulation experiments to validate if “neural networks can predict the performance of network topologies that have not been exposed to the model during training”. One point one should take extra care of is that many network topologies we study are very similar to many others, with shared motifs and links. These considerations extend beyond the scope of this paper.

Other potential improvements or future workComment 10: - The growth feedback examined in this paper comes from the effect of protein levels on the cell division rate (growth rate). However, the opposite effect can also occur; cell growth rates can affect the distribution of protein expression in cell populations. A good reference is Kheir Gouda et al., which is already on the list of references. These opposite effects should be described and discussed.

We agree that growth feedback is inherently complex and has many biological effects, and in our paper, we are using a simplified model to study the dominant factor of growth feedback. We have added the following paragraph in the Discussion section, which involves the opposite effect mentioned in the comment.

“In our paper, we consider dilution due to cell growth as the dominant factor of growth feedback. Here we compared the adaptive circuits under no-growth conditions and their ability to maintain their adaptive behaviors after dilution into a fresh medium, which mediated a significant dilution to the circuits. This is based on our previous work (Zhang et al. (2020)). However, growth feedback is inherently complex (Klumpp et al. (2009)). For instance, an increased growth rate can change protein synthesis rate (Hintsche and Klumpp (2013); Scott and Hwa (2023)), and cell growth rates can affect the distribution of protein expression in cell populations (Gouda et al. (2019)). In our paper, we concentrate on a simplified model with dilution, which we consider to have captured the dominant factor. The dynamic roles of the dilution and growth-affected production rate should be analogous, given that they both act as inhibitory factors arising from cell growth. Incorporating the impact of growth rate on protein synthesis into our model would offer a more comprehensive analysis, a task beyond the scope of this paper but presenting an intriguing opportunity for future research to address the complexities of growth feedback.”

Comment11: - It may be worth mentioning that growth feedback can lead to persistence, see PMID:27010473.

We have included this research as a citation.

Comment 12: - While some other networks (two-node) are discussed, it would be worth doing this analysis for all one- and two-node networks, perhaps controlled by small molecules added externally. If not here, then as a future plan.

We agree that this is an interesting idea for future studies.

Comment 13: - The manuscript analyzes the deterministic dynamics of a set of gene networks. However, gene expression is always stochastic, and gene circuits have been designed to control stochastic gene expression. For example, gene expression distributions can be reshaped, or even new peaks can appear, which would be worth mentioning, PMID: 30341217. The effect of growth feedback on stochastic gene expression and future perspectives of systematically studying this should be discussed.

We have added the following paragraph in the Discussion section to discuss the effects of noises and stochasticity. The research mentioned in the comment is also included.

“Our study focuses on scenarios where random noises are ignored. Realistically, gene circuits are subjected to diverse types of noise, which can complicate their predictable behavior and design. These noises can originate externally from a noisy input signal *I*, or intrinsically, directly affecting the circuit components. Further, these noises can be classified based on various mechanisms that cause them (Colin et al. (2017); Sartori and Tu (2011)). And with different mechanisms, each type of noise can be characterized by different attributes such as frequency, amplitude, and noise color. These variances can lead to different impacts on the circuits, potentially necessitating unique mechanisms or designs for the attenuation of each category (Sartori and Tu (2011); Qiao et al. (2019)). Given the extensive complexity and the need for thorough investigation, these noise-related challenges are beyond the scope of this paper and require a series of future studies.”